# Ferritin-mediated iron detoxification promotes hypothermia survival in *Caenorhabditis elegans* and murine neurons

Tina Pekec [1,2,7], Jarosław Lewandowski [3,7], Alicja A. Komur [3,7], Daria Sobańska [3], Yanwu Guo[4], Karolina Świtońska-Kurkowska [3], Jędrzej M. Małecki [4], Abhishek Anil Dubey[5], Wojciech Pokrzywa [5], Marcin Frankowski [6], Maciej Figiel [3] & Rafal Ciosk [3,4] ✉

How animals rewire cellular programs to survive cold is a fascinating problem with potential biomedical implications, ranging from emergency medicine to space travel. Studying a hibernation-like response in the free-living nematode *Caenorhabditis elegans*, we uncovered a regulatory axis that enhances the natural resistance of nematodes to severe cold. This axis involves conserved transcription factors, DAF-16/FoxO and PQM-1, which jointly promote cold survival by upregulating FTN-1, a protein related to mammalian ferritin heavy chain (FTH1). Moreover, we show that inducing expression of FTH1 also promotes cold survival of mammalian neurons, a cell type particularly sensitive to deterioration in hypothermia. Our findings in both animals and cells suggest that FTN-1/FTH1 facilitates cold survival by detoxifying ROS-generating iron species. We finally show that mimicking the effects of FTN-1/FTH1 with drugs protects neurons from cold-induced degeneration, opening a potential avenue to improved treatments of hypothermia.

Cold is a potentially lethal hazard. Nonetheless, hibernation is a widespread phenomenon used by animals to survive periods of low energy supply associated with cold[1–4]. Although humans do not hibernate, some primates do so[5], hinting that a hibernation-like state might, someday, be induced in humans, with fascinating medical repercussions[6,7]. Nowadays, cooling is used widely in organ preservation for transplantation. Therapeutic hypothermia is also applied, among others, during stroke or trauma, helping preserve functions of key organs, like the brain or heart[8,9]. Cellular responses to cold are also of interest for longevity research, as both poikilotherms (animals with fluctuating body temperature, like flies and fish) and homeotherms (like mice) live longer at lower temperatures[10,11]. Therefore,

understanding the molecular underpinnings of cold resistance has the potential to transform several areas of medicine.

The free-living nematode *C. elegans* populates temperate climates[12], indicating that these animals can survive spells of cold. In laboratories, *C. elegans* are typically cultivated between 20–25 °C, and a moderate temperature drop slows down but does not arrest these animals[13,14]. Deep cooling of *C. elegans*, i.e. to near-freezing temperatures, remains less studied. Exposing nematodes to 2–4 °C, after transferring them directly from 20–25 °C (which we refer to as "cold shock"), results in the death of most animals within 1 day of rewarming[15–17]. However, the lethal effects of cold shock can be prevented when animals are first subjected to a transient "cold

[1]Friedrich Miescher Institute for Biomedical Research, Maulbeerstrasse 66, 4058 Basel, Switzerland. [2]University of Basel, Faculty of Natural Sciences, Klingelbergstrasse 70, 3026 Basel, Switzerland. [3]Institute of Bioorganic Chemistry, Polish Academy of Sciences, Noskowskiego 12/14, 61-704, Poznań, Poland. [4]University of Oslo, Department of Biosciences, Blindernveien 31, Oslo, Norway. [5]Laboratory of Protein Metabolism, International Institute of Molecular and Cell Biology in Warsaw, Ks. Trojdena 4, 02-109, Warsaw, Poland. [6]Adam Mickiewicz University in Poznań, Faculty of Chemistry, Uniwersytetu Poznańskiego 8, 61-614, Poznań, Poland. [7]These authors contributed equally: Tina Pekec, Jarosław Lewandowski, Alicja Komur. ✉e-mail: rafal.ciosk@ibv.uio.no

acclimatization/adaptation" at an intermittent temperature of 10–15 °C[15,17]. Such cold-adapted nematodes can survive near-freezing temperatures for many days[15,17–19]. While in the cold, the nematodes stop aging, suggesting that they enter a hibernation-like state[17].

Among factors promoting *C. elegans* survival in near-freezing temperatures, we previously identified a ribonuclease, REGE-1, homologous to the human Regnase-1/MCPIP1[17,20]. In addition to ensuring cold resistance, REGE-1 promotes the accumulation of body fat, which depends on the degradation of mRNA encoding a conserved transcription factor, ETS-4[17]. Interestingly, previous studies showed that the loss of ETS-4 synergizes with the inhibition of insulin signaling in extending lifespan[21] and that the inhibition of the insulin pathway dramatically enhances cold survival[15,19]. Combined, these observations suggested that the cold survival-promoting function of REGE-1 could be related to the inhibition of the ETS-4/insulin signaling axis. In this work, we validate that hypothesis, dissect the underlying mechanism, and reveal that its main objective is the neutralization of harmful iron species. The connection between cold and iron toxicity is consistent with previous studies on mammalian cells[22–24]. We extend this analysis to mammalian neurons and describe a conserved mechanism that protects cells from cold damage by ferritin-mediated iron detoxification.

## Results

### Inhibition of ETS-4 improves *C. elegans* survival in the cold

Our initial studies of *C. elegans* "hibernation" identified the RNase REGE-1 as a factor promoting cold survival[17]. Studying REGE-1 in a different physiological context, the regulation of body fat, we showed that a key target of REGE-1 encodes a conserved transcription factor, ETS-4[17]. Thus, we asked whether overexpression of ETS-4, taking place in *rege-1(−)* mutants, is also responsible for their cold sensitivity. We tested that by incubating animals at 4 °C, henceforth simply the "cold" (for details on cold survival assay, see Fig. 1a). Indeed, we found that *rege-1(−); ets-4(−)* double mutants survived cold much better than *rege-1(−)* single mutants (Fig. 1b). Unexpectedly, however, the double mutants survived cold even better than wild type (Fig. 1b). Intrigued, we additionally examined the *ets-4(−)* single mutants and found that these mutants also survived cold much better than wild type (Fig. 1b). Thus, inhibiting ETS-4 is beneficial for cold survival irrespective of REGE-1. This observation was somewhat surprising as, in wild type, REGE-1 inhibits ETS-4 by degrading its mRNA[17]. However, we observed that, in wild type, both ETS-4 protein and *ets-4* mRNA were more abundant in the cold (Fig. 1c, d). Thus, an incomplete/inefficient degradation of *ets-4* mRNA in the cold could explain the improved survival of *ets-4(−)* mutants.

Because many hibernators burn accumulated fat to fuel survival in the cold, the ETS-4-mediated fat loss[17], and cold sensitivity observed here in *rege-1(−)* mutants, could be connected. Previously, we found that inhibiting ETS-4 restores body fat of *rege-1(−)* mutants to wild-type levels[17]. Here, we additionally examined the fat content of *ets-4(−)* single mutants. We observed that, at 20 °C, the fat content in these mutants was similar to wild type (Fig. 1e). We also examined fat levels after a few days in the cold and found that the *ets-4(−)* mutants had less fat than wild type (Fig. 1f). Thus, while the increased cold resistance of *ets-4(−)* mutants cannot be explained by the higher levels of energy stored in body fat, it is possible that higher consumption of fat in *ets-4(−)* mutants might be linked to improved cold survival.

### The enhanced cold survival requires both DAF-16 and PQM-1

ETS-4 was previously described to synergize with the insulin/IGF-1 signaling pathway in limiting the nematode lifespan[21]. Moreover, the lifespan extension seen in *ets-4(−)* mutants, as is the case with insulin pathway mutants, depends on the transcription factor DAF-16/ FOXO[21,25]. These and additional reports, that insulin pathway mutants display cold resistance depending on DAF-16[15,19], prompted us to

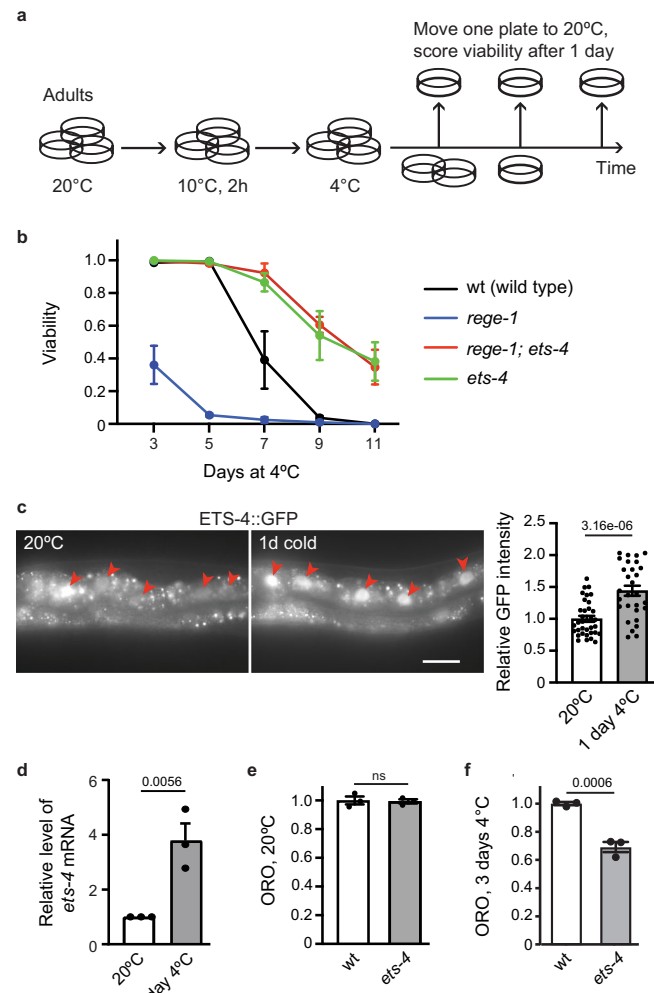

**Fig. 1 | ETS-4 is a negative regulator of cold survival. a** Representation of a typical cold-survival experiment. One-day-old adult nematodes, grown at 20 °C in multiple plates, were cold-adapted for 2 h at 10 °C and then shifted to 4 °C. Every few days, one plate was transferred back to 20 °C, and after 1 day of recovery, the animals were scored for viability. **b** Inactivation of *ets-4* increases cold survival. Animals of the indicated genotypes were cold-exposed as in 1a. Note that *ets-4(rrr16)* mutants, like *rege-1(rrr13); ets-4(rrr16)* double mutants, survived cold better than wild type (mutant alleles indicated in parentheses). Error bars represent the standard error of the mean (SEM), n = 3 independent experiments. 200–350 animals were scored per time point. **c** ETS-4 protein in upregulated in the cold. One-day-old adult *ets-4::GFP(rrr45)* nematodes were cold-exposed as in 1a. Representative images on the left show enhanced GFP fluorescence in the intestinal nuclei (arrowheads). Scale bar: 20 μm. The corresponding quantification is on the right. Error bars represent SEM, n = 1 independent experiment, 30–40 nuclei examined per condition. **d** Cold upregulates the *ets-4* mRNA. Wild-type nematodes were treated as in 1a. The level of *ets-4* mRNA, normalized to *act-1* mRNA, was measured by RT-qPCR and is expressed relative to control animals grown at 20 °C. Error bars represent SEM, n = 3 independent experiments. **e, f** Cold induces a faster loss of body fat in *ets-4* mutants than wild type. Wild type or *ets-4(rrr16)* nematodes were grown at 20 °C (**e**) or were incubated in the cold for 3 days (**f**), as in 1a. Body fat was quantified by staining with the lipophilic dye oil red O (ORO). Error bars represent SEM, n = 3 independent experiments. 10–15 animals (**e**) or 30 animals (**f**) were scored per replicate and strain. **c–f.** p values calculated using unpaired one-sided t-test; ns not significant. Source data are provided as a Source Data file.

examine the genetic relationship between *ets-4(−)* and the insulin pathway mutants in the context of cold resistance. First, using a loss-of-function allele of the insulin-like receptor, *daf-2(e1370)*[26], we observed that these mutants survived cold even better than *ets-4(−)* mutants (Fig. 2a). We then compared each single mutant with the

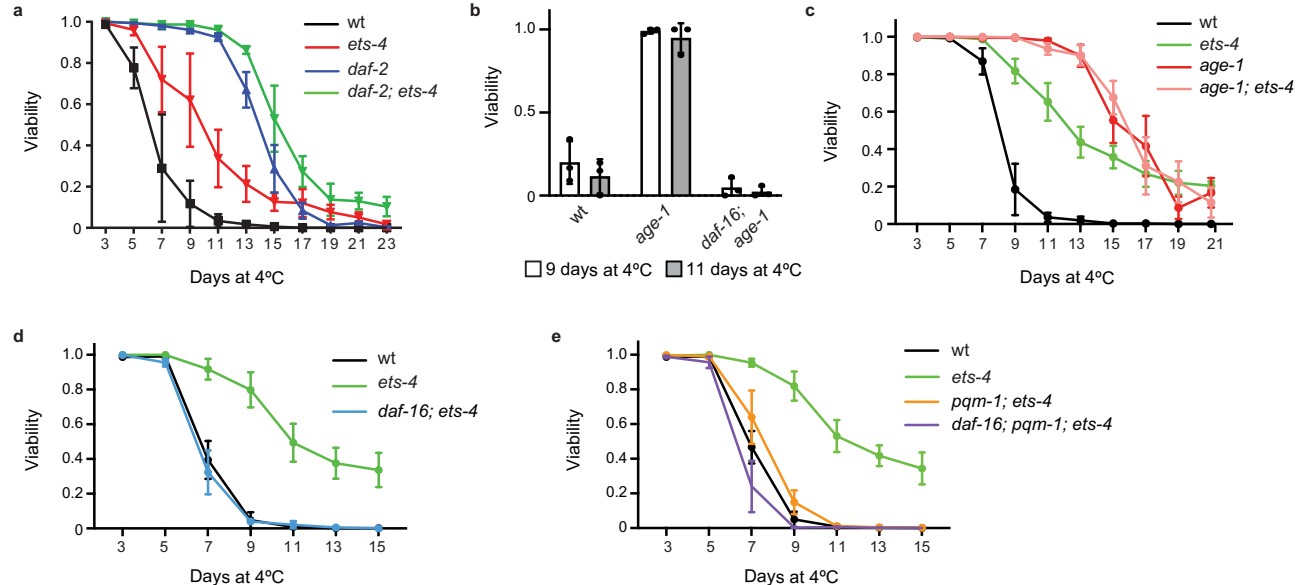

**Fig. 2 | The enhanced cold survival of *ets-4* mutants depends on DAF-16 and PQM-1. a**–**e** Survival of nematodes of the indicated genotypes, subjected to cold as in Fig. 1a. Mutant alleles are indicated in parentheses. Error bars represent SEM, "*n*" indicates the number of biologically independent replicates. **a** Inactivation of *ets-4* and *daf-2* increases cold survival. Cold resistance was compared between wild type (wt), *ets-4(rrr16)*, *daf-2(e1370)*, and *daf-2(e1370); ets-4(rrr16)* mutants. Note that *daf-2(e1370); ets-4(rrr16)* double mutant had only slightly improved cold resistance compared to *daf-2(e1370)* single mutant (*p* = 4.90E-06; Wilcoxon signed-rank test). *n* = 3, 324–668 animals scored per time point. **b** Increased cold resistance of *age-1* mutants depends on *daf-16*. Cold resistance was compared between wild type (wt), *age-1(hx546)*, and *daf-16(mu86); age-1(hx546)* mutants. *n* = 3, 200–300 animals scored per time point. **c** The cold resistance of *age-1*

mutants does not increase with an additional inactivation of *ets-4*. Cold resistance was compared between wild type (wt), *age-1(hx546)*, *ets-4(rrr16)* and *age-1(hx546); ets-4(rrr16)* mutants. *n* = 4, 350–500 animals scored per time point. **d** The increased cold resistance of *ets-4* mutants depends on *daf-16*. Cold resistance was compared between wild type (wt), *ets-4(rrr16)*, and *daf-16(mu86); ets-4(rrr16)* mutants. *n* = 4, 350–500 animals scored per time point. **e** The increased cold resistance of *ets-4* mutants also depends on *pqm-1*. Cold resistance was compared between wild type (wt), *ets-4(rrr16)*, *pqm-1(ok485); ets-4(rrr16)* and *daf-16(mu86); pqm-1(ok485); ets-4(rrr16)* mutants. Note that the triple mutants survived cold essentially like wild type, indicating that DAF-16 and PQM-1 promote cold survival in *ets-4(–)*, but not wild type animals. *n* = 4, 450–650 animals scored per time point. Source data are provided as a Source Data file.

double mutant, and observed that the *daf-2(e1370); ets-4(–)* double mutants survived cold only slightly better than the *daf-2(e1370)* single mutants (Fig. 2a; *p* = 4.9E-06). Since the effects of either single mutant do not simply add up, these observations suggest that the *daf-2(e1370)* and *ets-4(–)* mutants use, at least partly, overlapping mechanisms to promote cold survival.

That partial overlap suggests that ETS-4 may affect insulin signaling "downstream" from the DAF-2 receptor. The main components of the *C. elegans* insulin pathway include the phosphoinositide 3-kinase AGE-1/PI3K, which is why we also tested the genetic relationship between *age-1* and *ets-4* mutations. Using the *age-1(hx546)* allele, carrying a point mutation reducing the AGE-1 activity[27], we confirmed that also the *age-1* mutants survived cold better than the wild type, and that their improved survival depended on the transcription factor DAF-16/FOXO (Fig. 2b). Then, we examined the relationship between *age-1(hx546)* and *ets-4(–)* mutants. While the *age-1(hx546)* single mutants survived cold, expectedly, much better than the wild type, we observed no additional benefit of combining *age-1(hx546)* and *ets-4(–)* mutations (Fig. 2c). These observations suggest that AGE-1 could act in the same pathway as ETS-4, or converge on the same downstream effector(s). Thus, we also examined whether the enhanced cold survival of *ets-4(–)* mutants depends on DAF-16. Indeed, we found that removing DAF-16 completely suppressed the enhanced cold survival of *ets-4(–)* mutants (Fig. 2d). Reconciling all observations, we hypothesize that, in wild type, signals generated upon DAF-2 or ETS-4 activation converge on AGE-1, thus inhibiting DAF-16 and limiting cold resistance. Conversely, upon the inactivation of DAF-2 or ETS-4, DAF-16 activation results in improved cold resistance.

Recently, another transcription factor, PQM-1, was shown to complement DAF-16 in promoting the lifespan in DAF-2 deficient animals[28]. Although, in the intestinal cells, PQM-1 and DAF-16 nuclear

occupancy has been shown to be mutually exclusive, some evidence was provided supporting the synergistic roles of DAF-16 and PQM-1[28,29]. Therefore, we tested whether the loss of PQM-1 could have a similar effect on the cold survival of *ets-4(–)* mutants as the loss of DAF-16. Indeed, removing PQM-1 suppressed the enhanced cold survival of *ets-4(–)* mutants (Fig. 2e). Importantly, in an otherwise wild-type background, we observed no apparent effects on cold survival in either *pqm-1(–)* or *daf-16(–)* single mutants, nor in the *pqm-1(–); daf-16(–)* double mutants (Supplementary Fig. 1). Moreover, the survival of *daf-16(-); pqm-1(–); ets-4(–)* triple mutants was similar to wild type (Fig. 2e). Together, these observations argue for cold survival-promoting roles for DAF-16 and PQM-1, which become apparent under conditions that favor their activation, such as upon ETS-4 inactivation. Although either protein seems necessary for the full extent of cold protection in *ets-4(–)* mutants, we noticed that the cold survival of *daf-16(–); pqm-1(–); ets-4(–)* triple mutants was slightly (but significantly) reduced compared with the *pqm-1(–); ets-4(–)* double mutants (Fig. 2e; *p* = 0.01). Thus, while the bulk of cold protection mediated by DAF-16 and PQM-1 may reflect a shared function, these proteins could contribute to cold survival also by playing minor, independent roles.

## DAF-16 and PQM-1 are enriched in the gut nuclei in the cold

We hypothesized that DAF-16 and PQM-1 may facilitate cold survival by inducing the transcription of specific genes. Under normal growth conditions, DAF-16 remains inactive in the cytoplasm. However, when insulin signaling is inhibited, DAF-16 moves to the nucleus to activate target genes[30]. To test the nuclear accumulation of DAF-16 in our system, we attached (by CRISPR/Cas9 editing) a GFP-FLAG tag to the C-terminal end of the endogenous *daf-16* open reading frame (ORF) (see Methods) and examined the distribution of GFP-tagged DAF-16 (DAF-16::GFP) in wild type and *ets-4(–)* mutant nematodes. We

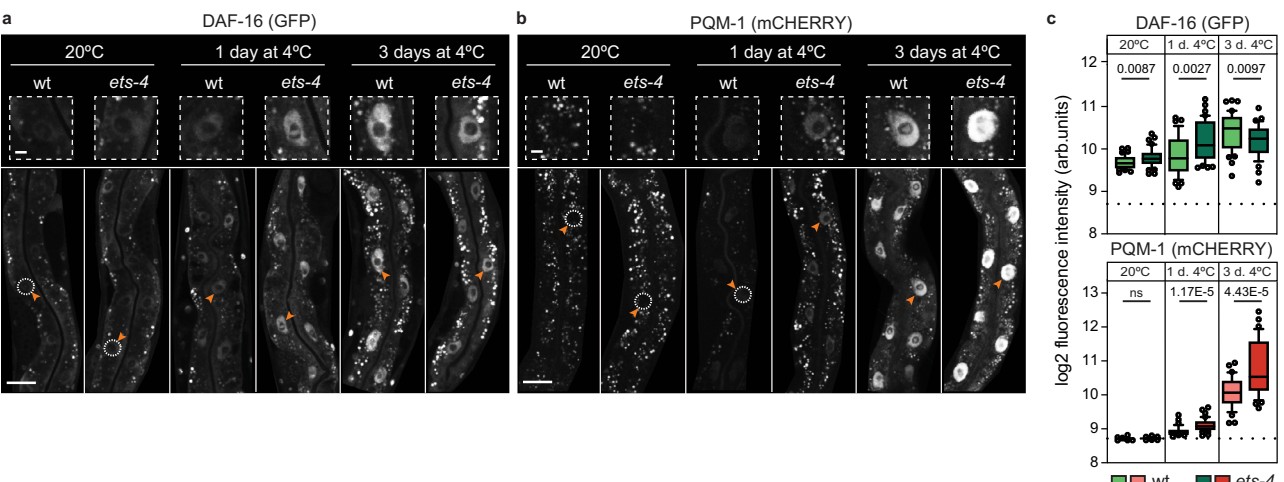

**Fig. 3 | DAF-16 and PQM-1 are enriched in the gut nuclei of cold-exposed nematodes. a** Representative confocal images showing nuclear accumulation of DAF-16. Wild type or *ets-4(rrr16)* mutants, both containing endogenously GFP-tagged DAF-16 (allele *daf-16(syb707)*), were sampled at the indicated times and temperatures, as in Fig. 1a, and the GFP fluorescence was imaged (quantified in **c**, left). Arrowheads point to representative gut nuclei (demarcated with dashed circles when displaying little or no fluorescence), which are enlarged in the insets above. Size bars, here and in **b**: 25 μm (small magnification) and 5 μm (large magnification). **b** Representative confocal images showing nuclear accumulation of PQM-1. Wild type or *ets-4(rrr16)* mutants, both containing endogenously mCherry-tagged PQM-1 (allele *pqm-1(syb432)*), were sampled as above, and mCherry fluorescence was imaged (quantified in **c**, right). Arrowheads point to representative gut nuclei, enlarged in the insets. **c** Quantifications of the nuclear fluorescence, corresponding to **a** (left) and **b** (right). Each data point represents log$_2$-transformed mean nuclear intensity per animal. The dotted line represents the average background within each experiment. Box plot: center line, median; box limits, upper and lower quartiles; whiskers, tenth to 90th percentile; points, outliers. Data from three biologically independent replicates, 10 to 15 animals scored per replicate. *p* values calculated using unpaired two-sided *t*-test; ns not significant. Source data are provided as a Source Data file.

observed little nuclear signal at 20 °C, with a minimal increase in the absence of ETS-4 (Fig. 3a, c). After 1 or 3 days at 4 °C, however, we observed a significant increase in the nuclear DAF-16::GFP (Fig. 3a, c); this increase appeared to be posttranscriptional, as *daf-16* mRNA levels remained constant between 20 and 4 °C (Supplementary Fig. 2a). Although the nuclear DAF-16::GFP signal appeared slightly stronger in *ets-4(−)* mutants at day one in the cold, this was no longer true at day 3 (Fig. 3a, c). Thus, although the nuclear enrichment of DAF-16 is consistent with its ability to potentiate cold resistance, that enrichment is, apparently, insufficient, as it only enhances cold survival in *ets-4(−)* mutants but not in wild type (Fig. 2a).

Next, we performed a similar analysis on PQM-1, fusing (by CRISPR/Cas9 editing) an mCHERRY-MYC tag to the C-terminal end of the endogenous *pqm-1* ORF (see Methods). We detected little if any nuclear PQM-1::mCHERRY at 20 °C in either wild type or *ets-4(−)* mutants (Fig. 3b, c), agreeing with the previously reported expression patterns[28,31–33]. By contrast, after 1 day at 4 °C, we began detecting the nuclear PQM-1::mCHERRY signal in wild-type nematodes and a slightly stronger signal in the nuclei of *ets-4(−)* mutants (Fig. 3b, c); this increase may be transcriptional, as *pqm-1* mRNA levels were higher at 4 °C than 20 °C (Supplementary Fig. 2b). After three days at 4 °C, the PQM-1::mCHERRY nuclear signal increased even further and, at this point, *ets-4(-)* mutants displayed significantly higher signal than wild type (Fig. 3b, c). Thus, in contrast to the standard cultivation temperature, where DAF-16 and PQM-1 localize to the nucleus in a mutually exclusive manner[28], DAF-16 and PQM-1 coexist in the nucleus in the cold.

### Identification of *ftn-1* as a candidate gene promoting cold survival

The above observations are compatible with a scenario where, upon ETS-4 inactivation, DAF-16 and PQM-1 co-regulate transcription of cold survival-promoting gene(s). To test this hypothesis, we undertook a functional genomic approach. First, we compared gene expression (by RNA-seq) between *ets-4(−)* and wild-type animals incubated at 4 °C. Then, by comparing *pqm-1(−); ets-4(−)* or *daf-16(−); ets-4(−)* double

mutants to the *ets-4(−)* single mutant, we identified genes, whose expression in the *ets-4(−)* mutant depends on PQM-1 and/or DAF-16 (Supplementary Data 1). To illustrate this, we prepared an integrative heat map, using all 4 °C samples with replicates. Focusing on changes between the strains, we observed three distinct gene clusters (Fig. 4a). Cluster 1 (red) includes genes upregulated in the cold in *ets-4(−)* mutants (compared to wild type), which either do not change or go down upon the additional inhibition of *daf-16* or *pqm-1*. Cluster 2 (green) includes genes upregulated across all conditions. Finally, the smallest cluster 3 (blue), includes genes downregulated in *ets-4(−)* mutants, which either do not change or go up, upon the additional inhibition of *daf-16* or *pqm-1*. With this analysis, we observed that many changes in gene expression upon the loss of ETS-4 were reverted upon the additional loss of either DAF-16 or PQM-1, supporting a functional relationship between DAF-16 and PQM-1. Taking advantage of the ENCODE database, which reports genome-wide chromatin association of many transcription factors[34], we examined the potential binding of DAF-16 and PQM-1 around the transcription start sites (TSS) of genes in each cluster of the heat map. Even though the ENCODE data comes from experiments performed at standard growth conditions, we decided to use it as an approximation and observed that genes whose expression in *ets-4(−)* mutants depends on DAF-16 or PQM-1 (i.e., genes in clusters 1 and 3), appear to be enriched for TSS-proximal binding sites for both transcription factors (Fig. 4a). The same enrichment was not seen for the cluster 2 genes, whose expression is apparently unrelated to DAF-16 or PQM-1 (Fig. 4a). The possible connection between clusters 1 and 3, and the association with DAF-16 or PQM-1, was statistically significant for PQM-1 but not DAF-16 (Fig. 4a). Nevertheless, by analyzing transcription factor binding motifs enriched within each gene cluster, we observed both PQM-1-like and DAF-16-like motifs enriched within the cluster 1 genes (Supplementary Fig. 3a, b[28]), which made us focus on this group of genes.

To identify candidate genes, whose DAF-16 and PQM-1 dependent activation promotes cold survival, we first selected genes upregulated (in both biological replicates), at least twofold, in *ets-4(−)* mutants compared to wild type (after 1 day at 4 °C). Second, we intersected

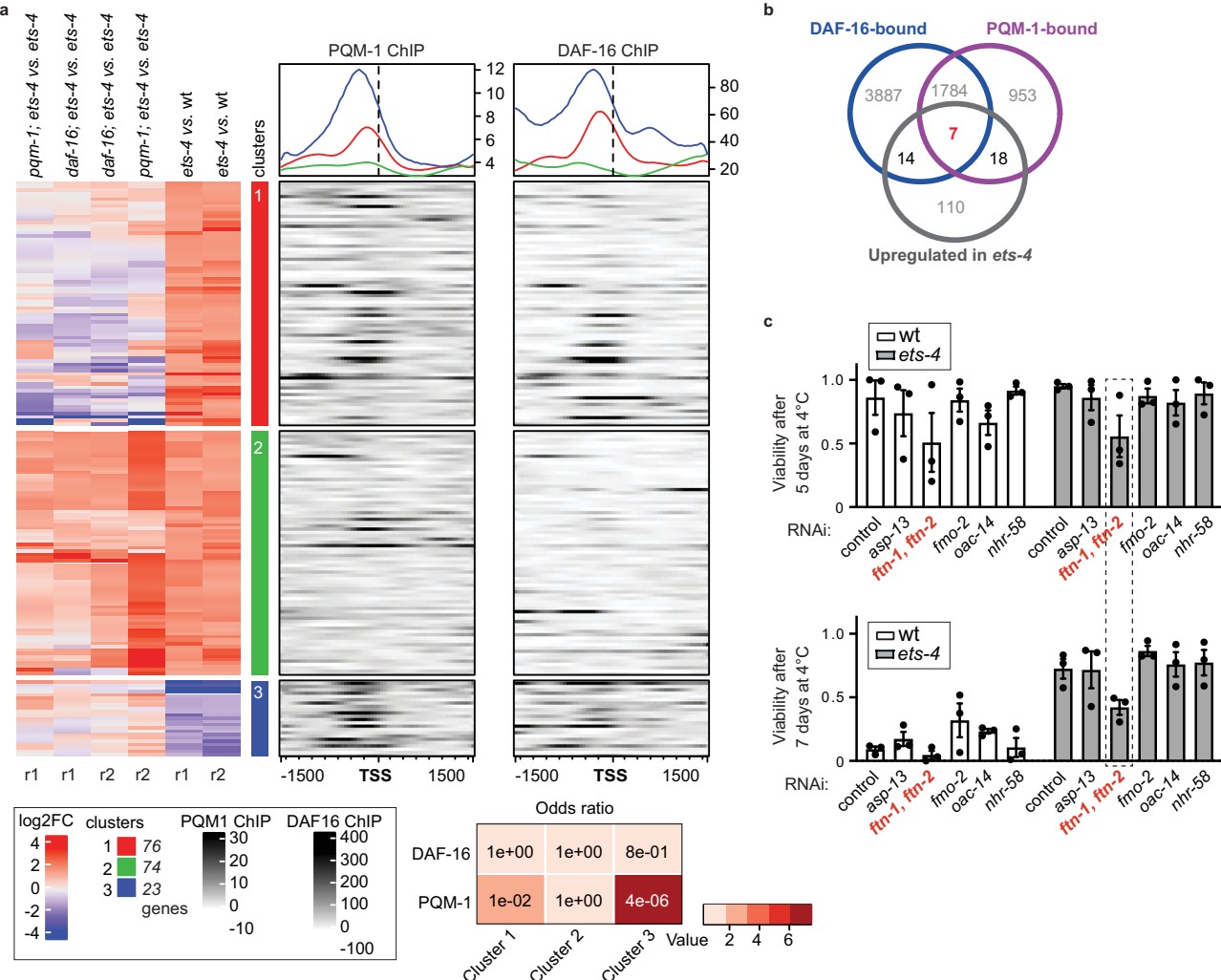

**Fig. 4 | Identification of *ftn-1* as a candidate gene promoting cold survival.**
**a** Heat map and clustering of genes differentially expressed in the cold. RNA-seq transcriptome analysis was performed on animals of the indicated genotypes: wt, *ets-4(rrr16)*, *daf-16(mu86); ets-4(rrr16)*, and *pqm-1(ok485); ets-4(rrr16)*. The animals, treated as in Fig. 1a, were collected on day 1 at 4 °C. Left: Integrative heat map showing log₂ fold changes in gene expression between the indicated strains. "r1 and r2" indicate biological replicates. Each line represents one gene. Automated clustering showed three distinct clusters: cluster 1 (red), cluster 2 (green), and cluster 3 (blue). Right: binding of DAF-16 and PQM-1 around the transcription start sites (TSSs) of genes shown on the left, using the ChIP ENCODE data[34]. The line graphs above (colored according to the clusters) illustrate enrichments for DAF-16 or PQM-1 binding, within each cluster, around the TSS. The heat map below shows a hypergeometric test of overlaps between three clusters of genes and PQM-1 or DAF-

16 targeted genes. Color-coded *p* values are shown. **b** Diagrams showing the overlap between the indicated sets of genes. Gray circle: genes upregulated more than twofold in *ets-4(rrr16)* mutants compared to wt at day 1 at 4 °C. Blue circle: genes whose promoters are bound by DAF-16. Magenta: genes whose promoters are bound by PQM-1 (according to ref. 28). Note that seven genes (*asp-13, ftn-1, fmo-2, oac-14, nhr-58, cpt-4*, and *pals-37*), whose promoters were bound (at 20 °C) by both DAF-16 and PQM-1, were reproducibly upregulated in the absence of ETS-4. **c** A joint depletion of *ftn-1* and *ftn-2* reduces cold survival. Wild type (wt) and *ets-4(rrr16)* mutants were RNAi-depleted for candidate genes from **b** (the depletion of *ftn-1* and *ftn-2*, in *ets-4* animals, is indicated by the stippled box), and tested for cold resistance, as in Fig. 1a. Error bars represent SEM, *n* = 3 independent experiments. 200–350 animals scored per time point. Source data are provided as a Source Data file.

these genes with those whose promoters associate with either DAF-16 or PQM-1, according to the confident binding sites from ref. 28. This analysis yielded seven genes (*asp-13, ftn-1, fmo-2, oac-14, nhr-58, cpt-4,* and *pals-37*) that were reproducibly upregulated in *ets-4*(−) mutants and whose promoters may associate with both PQM-1 and DAF-16 (Fig. 4b). If these genes were relevant for the enhanced cold survival, their inhibition would be expected to impede cold survival of *ets-4*(−) mutants. Testing this, we observed that RNAi-mediated depletion of *ftn-1* (encoding one of two nematode ferritins), reproducibly compromised cold survival (Fig. 4c; *pal-37* and *cpt-4* were tested subsequently, but their depletion did not have consistent effects on cold survival). Contrary to expectations, we observed that effect not only in *ets-4*(−) mutants but also in wild type (Fig. 4c). However, the RNAi construct used for targeting *ftn-1* is predicted to also target the other *C.*

*elegans* ferritin, *ftn-2*. While *ftn-2* is expressed constitutively, the expression of *ftn-1* is highly dynamic[35,36]. Thus, a baseline expression of *ftn-2* and/or *ftn-1* may contribute to the wild-type cold survival, while the additional induction of *ftn-1* could be responsible for the enhanced cold survival of *ets-4*(−) mutants. To test this hypothesis, we examined, by RT-qPCR, the expression of *ftn-1* and *ftn-2* in various mutants.

**FTN-1 induction by PQM-1 and DAF-16 explains the enhanced survival of *ets-4* mutants**
Consistent with the above scenario, we observed an upregulation of *ftn-1* (but not *ftn-2*) in wild type (weaker) and *ets-4* mutants (stronger) in the cold (Fig. 5a, b). Moreover, in agreement with our RNA profiling data, the *ftn-1* induction depended on both DAF-16 and PQM-1, whereas the expression of *ftn-2* did not (Fig. 5a, b). While a joint

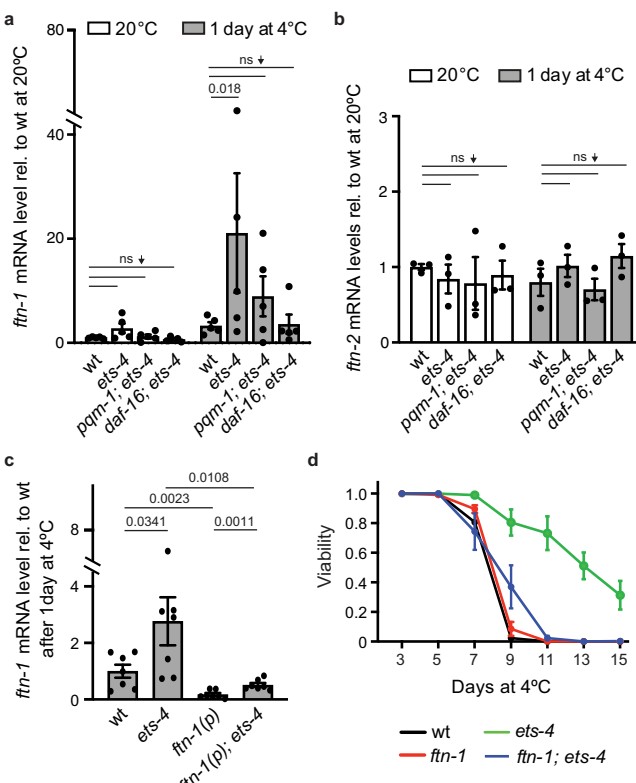

**Fig. 5 | FTN-1 upregulation promotes cold survival. a** DAF-16 and PQM-1 are required for the elevated expression of *ftn-1* mRNA in cold-treated *ets-4* nematodes. Wild type (wt), *ets-4(rrr16)*, *pqm-1(ok485); ets-4(rrr16)* and *daf-16(mu86); ets-4(rrr16)* mutants were cold-exposed, as in Fig. 1a. The level of *ftn-1* mRNA (normalized to *act-1* mRNA) was measured by RT-qPCR and expressed relative to wt animals at 20 °C. Error bars represent SEM, *n* = 5 independent experiments. *p* values calculated using two-way ANOVA with Tukey's multiple comparison test; ns not significant. **b** DAF-16 and PQM-1 do not regulate the expression of *ftn-2*. The animals were collected and analyzed for *ftn-2* mRNA level, similar as in **a**. *n* = 3 independent experiments. **c** Deletion of a putative PQM-1 binding site in *ftn-1* promoter abolishes the induction of *ftn-1* in the cold. Wild type (wt), *ets-4(rrr16)*, *ftn-1(p)*, and *ftn-1(p); ets-4(rrr16)* mutants were cold-exposed for 1 day, as in Fig. 1a. *ftn-1(p)* denotes the *ftn-1* allele *syb4641*, with putative PQM-1 biding site deleted from the promoter region. The level of *ftn-1* mRNA (normalized to *act-1* mRNA) was measured by RT-qPCR and expressed relative to cold-treated wt animals. Error bars represent SEM, *n* = 7 independent experiments. *p* values calculated using an unpaired one-sided *t*-test. **d** Inactivation of *ftn-1* abolishes the increased cold resistance of *ets-4* mutants. Cold resistance was compared between wild type (wt), *ftn-1(ok3625)*, *ets-4(rrr16)*, and *ftn-1(ok3625); ets-4(rrr16)* mutants, as in Fig. 1a. Error bars represent SEM, *n* = 3 independent experiments. 250–400 animals scored per time point. Source data are provided as a Source Data file.

induction by both TFs has not been reported before, the selective induction of *ftn-1* by DAF-16 and PQM-1 is consistent with previous reports that described, in different physiological contexts, DAF-16 or PQM-1-mediated upregulation of *ftn-1*[32,37]. Thus, expectedly, when examining the ENCODE data[34], we observed the association of DAF-16 and PQM-1 with the *ftn-1*, but not *ftn-2*, promoter region (Supplementary Fig. 3c). While PQM-1 associates only with the *ftn-1* promoter region, DAF-16 appears to bind with the first exon and within the promoter region. Interestingly, within the *ftn-1* promoter, DAF-16 and PQM-1 associate with the same region. Upon closer inspection, we noticed that this region contains a predicted PQM-1 binding site (Supplementary Fig. 3c)[28]. To test if this site is important for *ftn-1* induction, we created a strain in which this sequence was deleted from the promoter (pΔ *ftn-1*). Importantly, this deletion prevented *ftn-1* mRNA induction in the cold, in both wild type and *ets-4(−)* mutants

(Fig. 5c). Since FTN-1 is expressed in the intestine[36], i.e., the tissue where ETS-4, DAF-16, and PQM-1 are all expressed in the cold (Figs. 1c, 3a, b), our combined data support the scenario where *ftn-1* (but not *ftn-2*) is induced in cold-exposed *ets-4(−)* mutants jointly by PQM-1 and DAF-16. If the induction of *ftn-1* were responsible for the enhanced cold survival of *ets-4(-)* mutants, inhibiting *ftn-1* would be expected to impact cold survival of *ets-4(−)* mutants but not wild-type animals. We tested that using an existing loss-of-function allele, *ftn-1(ok3625)*[38,39]. Indeed, while the cold survival of *ftn-1(−)* single mutants was indistinguishable from wild type, the *ftn-1* inactivation abolished the enhanced cold survival of *ets-4(−)* mutants (Fig. 5d).

## FTN-1 promotes cold survival through its ferroxidase activity

Mammalian ferritin consists of multiple heavy and light subunits (FTH and FTL) that form nanocages storing thousands of iron atoms[40]. The *C. elegans* ferritins, FTN-1 and −2, are more similar to FTH than FTL[41]. Under standard growth conditions, when FTN-1 is expressed at low levels[36], FTN-2 is chiefly responsible for the storage of body iron[39]. To examine that in the cold, when FTN-1 is induced, we analyzed iron content in extracts from wild type or mutant animals, using size exclusion chromatography–inductively coupled plasma-mass spectrometry (SEC-ICP-MS)[39]. The bulk of iron was associated with a protein complex similar in size to the horse ferritin standard (~440 kDa), representing the *bona fide C. elegans* ferritin (annotated as Peak 2 in presented chromatograms). We found that, in cold-treated animals, the levels of total and ferritin-associated iron were independent of FTN-1, even in *ets-4(−)* mutants that express FTN-1 (Fig. 6a and Supplementary Fig. 4a). However, similar to standard temperature[39], they depended on FTN-2 (Fig. 6b and Supplementary Fig. 4a). Especially Peak 2 was nearly abolished in *ftn-2(−)* mutants, confirming that this peak represents ferritin-associated iron. Thus, the induction of FTN-1 appears to have little effect on the pool of ferritin-stored iron.

Iron is present in cells in both oxidized $Fe^{3+}$/ferric(III) and reduced $Fe^{2+}$/ferrous(II) forms. Excess of $Fe^{2+}$ is potentially harmful because, in the so-called Fenton reaction, it catalyzes the formation of reactive oxygen species (ROS)[42,43]. Notably, both FTN-1 and −2 contain predicted ferroxidase active sites. In particular, the residues Glu-58 and His-61, corresponding to Glu-63 and His-66 in human FTH1 (Fig. 6c and Supplementary Fig. 4b, c), which in homologous proteins mediate the $Fe^{2+}$-to-$Fe^{3+}$ conversion[44]. This strongly suggests that both FTN-1 and −2 are ferroxidase-active. Accordingly, ferritin-deficient mutants display an elevated ratio of $Fe^{2+}/Fe^{3+}$ during aging[39]. Although the individual impact of FTN-1 on the $Fe^{2+}/Fe^{3+}$ balance was not examined, the overexpression of *ftn-1* was reported to have antioxidant effects[45]. Thus, we wondered if the ferroxidase activity of FTN-1 is beneficial for enhanced cold survival. To show that FTN-1 and −2 are indeed ferroxidase-active, we expressed recombinant, untagged FTN-1 and −2 in *E. coli* and purified them to apparent homogeneity (Supplementary Fig. 4d). In addition, we expressed and purified the putative ferroxidase inactive versions of FTN-1 and −2, by introducing the E58K/H61G double-mutation (FeOx-mut) to mimic the sequence of the human ferritin light chain (FTL) (Fig. 6c), which is ferroxidase inactive[46]. Recombinant FTN-1 and −2, both wt and mutated, formed high molecular weight complexes, suggesting correct folding and formation of ferritin cages (Supplementary Fig. 4d). Importantly, the wild-type FTN-1 and −2 proved to be ferroxidase active, the E58K/H61G mutants were not (Fig. 6d). Additionally, we tested the ability of recombinant FTNs to bind iron in conditions when ferrous ions are the only iron source. We found that the initial velocity (rate) of iron binding was reduced in E58K/H61G-mutated FTNs compared to wild-type proteins (Fig. 6e). A simple interpretation of this result is that, at least in vitro, the rate of binding reflects the speed with which the ferric iron is generated from ferrous ions in the presence (fast, catalyzed) or absence (slow, spontaneous oxidation) of the FTN ferroxidase activity. Importantly, and in agreement with similar results obtained with human ferritins[44,46], the

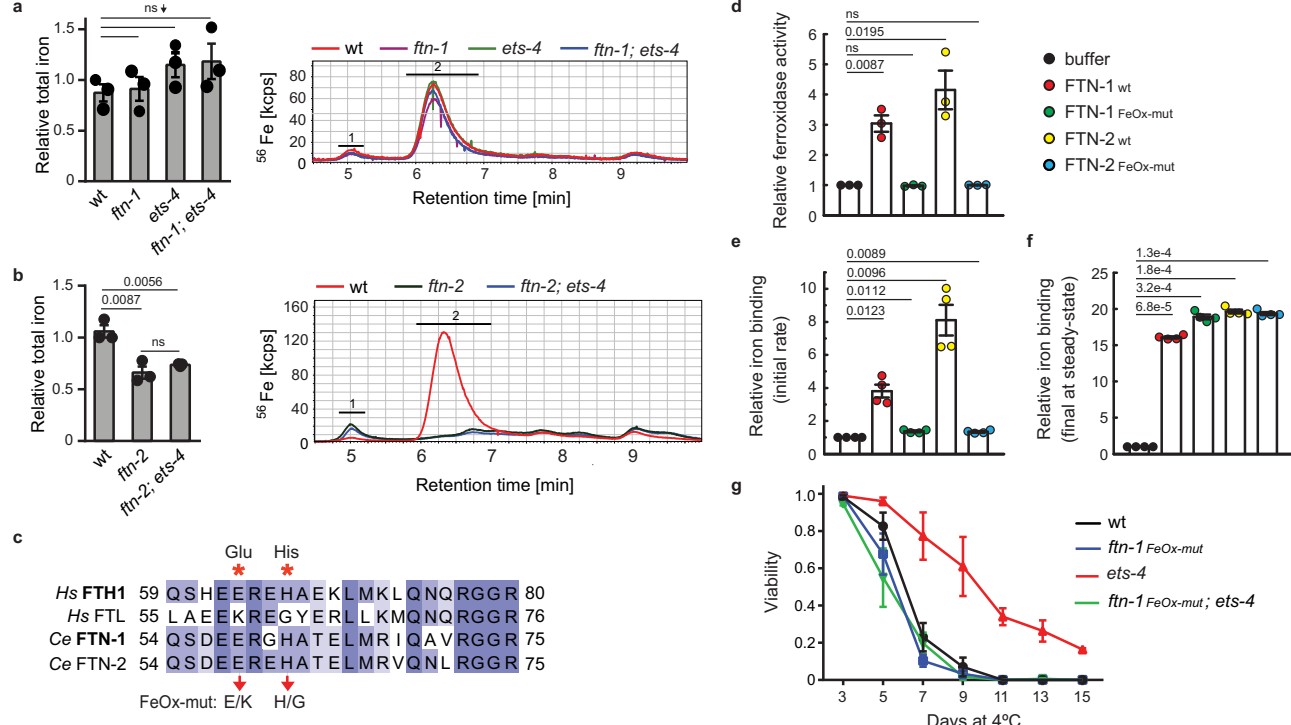

**Fig. 6 | FTN-1 promotes cold survival via its ferroxidase activity. a** Total body iron is independent of FTN-1. One-day-old adults: wt, *ftn-1(ok3625)*, *ets-4(rrr16)*, and *ftn-1(ok3625); ets-4(rrr16)*, were cold-exposed (3 days). Total iron in extracts was measured by SEC-ICP-MS and normalized to wt. Right: chromatograms, left: quantitation of area under curves. Peak 1; high molecular weight-associated iron. Peak 2; ferritin-associated iron. *n* = 3 independent experiments. **b** Total body iron depends on FTN-2. One-day-old adults: wt, *ftn-2(ok404)*, and *ftn-2(ok404)*; *ets-4(rrr16)*, were cold-exposed (3 days). Total iron in extracts was measured as in **a**. *n* = 3 independent experiments. **c** Partial sequence alignment of ferritin homologs. Human (Hs): ferritin heavy (FTH1; NP_002023.2) and light (FTL; NP_000137.2) chains. *C. elegans* (Ce): FTN-1 (NP_504944.2) and FTN-2 (NP_491198.1). Red asterisks indicate amino acids mutated (E58K/H61G) in ferroxidase-deficient FTNs (FeOx-mut). **d** The FeOX-mut mutations abrogate the ferroxidase activity of FTNs. Ferroxidase activity of recombinant FTN-1 or −2, either wt or FeOx-mut, was tested on

ferrous ammonium sulfate (FAS) relative to buffer (no FTN). *n* = 3 independent experiments. **e** The FeOX-mut mutations reduce the initial velocity of iron binding. The initial rate of iron binding after mixing FAS with recombinant FTNs, as in **d**, was measured relative to buffer. *n* = 4 independent experiments. **f** The FeOX-mut mutations do not impact a steady-state iron binding. Recombinant FTNs, as in **d**, were incubated with FAS for 1 h. The amount of soluble (FTN-bound) iron was measured relative to the buffer. *n* = 4 independent experiments. **g** The FeOx-mut mutations in *ftn-1* abolish increased cold resistance of *ets-4* mutants. Cold survival of wt, *ftn-1(FeOx-mut)*, *ets-4(rrr16)* and *ftn-1(FeOx-mut); ets-4(rrr16)* animals was determined as in Fig. 1a. (*FeOx-mut* indicates the *ftn-1(syb2550)* allele). *n* = 3 independent experiments. 261–396 animals scored per time point. Statistical analysis: Error bars: SEM; *p* values calculated using unpaired two-sided *t*-test (in **a**, **b**) or paired one-sided *t*-test (in **d**–**f**); ns not significant. Source data are provided as a Source Data file.

ferroxidase activity of FTNs does not seem to be essential for the iron-binding per se because, after prolonged incubation (at steady-state), FTNs were able to bind a comparable amount of iron, regardless of whether they were ferroxidase-active or not (Fig. 6f).

Finally, to test whether the ferroxidase activity of FTN-1 is important for cold survival in vivo, we modified the endogenous *ftn-1* locus (introducing the E58K/H61G mutation by CRISPR/Cas9 editing), so that the mutant animals produce only a ferroxidase-inactive FTN-1 (FeOx-mut). Crucially, we found that the inactivation of the FTN-1 ferroxidase activity completely abolished the enhanced cold survival of *ets-4(−)* animals (Fig. 6g). Since commercially available anti-ferritin antibodies failed to detect FTN-1 and/or −2, we employed mass spectrometry to detect FTN-1- and FTN-2-specific peptides in animal extracts. Using this semi-quantitative approach we detected comparable amounts of FTN-1 and −2, in both *ets-4(−)* and *ftn-1(FeOx-mut); ets-4* mutants, indicating that E58K/H61G mutation has little impact on expression of both ferritins (Supplementary Fig. 4e). Notably, FTN-1 accounted only for ~10% of total ferritin level in both mutant animals (Supplementary Fig. 4e), which agrees with the minor contribution of FTN-1 to ferritin-associated iron stored in *C. elegans* (Fig. 6a). Thus, because FTN-1 does not seem to play a major role in iron storage but its ferroxidase activity is essential for enhanced cold survival, our results suggest that FTN-1 facilitates cold survival through the detoxification of ferrous iron.

## Overproduction of FTN-1 is sufficient for the enhanced cold survival

Thus far, we have shown that FTN-1, when expressed in the absence of ETS-4, gives nematodes an advantage in surviving cold. To test whether FTN-1 may do that in an otherwise wild-type background, we created (using Mos1-mediated Single Copy Insertion, MosSCI;[47]) strains overexpressing *ftn-1* from two different, robust promoters; *dpy-30* and *vit-5*. Importantly, we found that both strains survived cold much better than wild type (Fig. 7a; for the levels of *ftn-1* mRNA overproduced from the *vit-5* promoter, see Supplementary Fig. 5a). By SEC-ICP-MS, we observed no changes in the levels of total and ferritin-associated iron in *ftn-1* overexpressing strains (Fig. 7b and Supplementary Fig. 5b, c), consistent with FTN-1 being important for iron detoxification but not storage. Because we test animal survival following a multi-day incubation in the cold, the beneficial effect of FTN-1 overexpression could reflect a more general function of FTN-1 in counteracting ageing. Although FTN-1 was reported to have no major role in the lifespan of animals grown at standard temperature[36,45], we additionally compared the lifespan of wild-type or FTN-1 overexpressing animals as they emerged from a several-day incubation in the cold. In contrast to the strong cold survival-promoting effect of FTN-1, we observed only a small lifespan extension in animals that were surviving for the longest time (Supplementary Fig. 5d). Whether the benefit that those long-lived animals

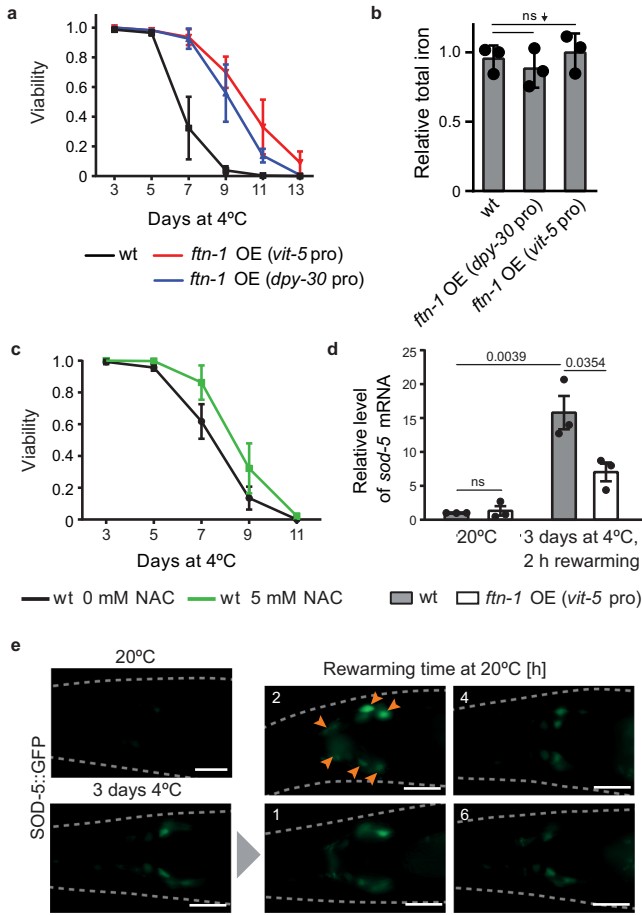

**Fig. 7 | FTN-1 overexpression is sufficient for enhanced cold survival. a** FTN-1 overexpression (OE) increases cold resistance. Cold survival was compared between wt, *ftn-1* OE *(vit-5* pro*)*, and *ftn-1* OE *(dpy-30* pro*)* animals, as in Fig. 1a *(dpy-30* pro *(sybSi67)* and *vit-5* pro *(sybSi72)* indicate OE lines, with *ftn-1* gene under control of *dpy-30* or *vit-5* promoter). Error bars represent SEM, *n* = 3 independent experiments. 232–307 animals scored per time point. **b** Total iron is unaffected by FTN-1 overexpression. One-day-old animals, as in **a**, were grown at 20 °C, and total iron in extracts was measured and analyzed as in Fig. 6a, with values normalized to wild type. Error bars indicate SEM, *n* = 3 independent experiments. ns not significant (unpaired two-sided *t*-test). **c** Antioxidant treatment improves cold survival of wild-type animals. Cold survival was compared between wt incubated in the absence or presence of *N*-acetylcysteine (NAC, 5 mM), as in Fig. 1a. Error bars represent SEM, *n* = 4 independent experiments. 452–514 animals scored per time point. **d** FTN-1 overexpression reduces *sod-5* mRNA elevated during rewarming of cold-exposed nematodes. One-day-old animals, wt or *ftn-1* OE *(vit-5* pro*)*, were collected at 20 °C or after 3 days at 4 °C with 2 h of rewarming at 20 °C. The levels of *sod-5* mRNA (normalized to *act-1* mRNA) were measured by RT-qPCR and expressed relative to wt at 20 °C. Error bars represent SEM, *n* = 3 independent experiments. *p* values calculated using unpaired two-sided *t*-test; ns not significant. **e** The expression of SOD-5::GFP peaks during rewarming. Representative fluorescence micrographs showing the expression of SOD-5::GFP fusion protein in the head region in 1-day-old adults subjected to cold. The animals were cold-exposed for 3 days, transferred back to 20 °C, and examined during rewarming in a time-course fashion. Representative images were taken at the indicated times and temperatures. Animals are outlined with gray, dashed lines. The experiment was repeated two times, with similar results. Scale bar: 20 μm. Source data are provided as a Source Data file.

receive from FTN-1 is the same as in the animals subjected to cold remains to be tested. Nonetheless, FTN-1 appears to be more important for cold survival than for lifespan extension.

The ferroxidase activity of FTN-1 is expected to lower the levels of ROS-generating Fe(II), implying that cold-treated nematodes experience increased levels of ROS. If so, treating animals with antioxidants

might be expected to improve cold survival. Although antioxidants are relatively unstable, and our protocols involve long incubation times, we tested that by adding to culture plates *N*-acetylcysteine (NAC), which was shown to provide significant protection upon oxidative stress treatment[48]. In agreement with our hypothesis, we observed a small but significant extension of cold survival (Fig. 7c; *p* = 0.0005). We then attempted to monitor ROS levels in cold-treated animals by quantifying the fluorescence intensity of DHE (dihydroethidium)-stained nematodes by following a published ROS-detection protocol[49]. We did observe the expected trend, i.e., less ROS in *ets-4* mutants than wt, and comparable to wt levels of ROS in *ftn-1* or *ftn-1; ets-4* double mutants (Supplementary Fig. 5e). However, the variation was large, and so the *p* value between wt and *ets-4* (0.0547; *t*-test) was just above the significance threshold (0.05). Thus, we sought another factor whose induction could be used as a proxy for ROS detection. Specific enzymes, called superoxide dismutases (SODs), function at the front line of cellular defense against ROS[50]. There are five SODs in *C. elegans* and, examining their expression in cold-treated animals, we noticed a consistent increase in the *sod-5* mRNA (Fig. 7d). To understand the dynamics of *sod-5* activation, we examined the expression of GFP-tagged SOD-5[51]; the fusion protein is expressed mainly in neurons (Fig. 7e). Following SOD-5::GFP signal in live animals, we observed a strong, but transient increase of SOD-5::GFP during rewarming (Fig. 7e; note the elevated signal around 2 h into rewarming). Focusing thus on this time point, we tested whether the overexpression of *ftn-1* impacts *sod-5* activation. Indeed, we observed that the levels of *sod-5* mRNA were significantly lower in the *ftn-1* overexpressing strain than wild type (Fig. 7d).

All observations combined, a picture emerges where FTN-1, through its iron(II)-detoxifying activity, protects animals from the cold by reducing the levels of Fe(II)-catalyzed ROS. According to this model, animals subjected to cold experience an increase in Fe(II) iron. Detection of specific iron forms is not trivial, and our attempts to detect specifically Fe(II) in *C. elegans* were unsuccessful. Thus, assuming some level of conservation in cellular responses to cold, we decided to investigate that in mammalian cells, where Fe(II) detection is more robust and potential findings more directly applicable to human hypothermia.

**Iron management plays a key role in neuronal resistance to cold**

Since the main clinical benefit of deep cooling is the preservation of neuronal functions, we decided to examine Fe(II) in neurons, where our observations may be of clinical relevance. For convenience, we chose to study murine neurons. To generate them, we differentiated primary neural stem cells, collected from early mouse embryos, into noradrenergic-like neurons (henceforth "neurons"), which affect numerous physiological functions, generally preparing the body for action. To examine their cold resistance, neurons (cultivated at the physiological temperature of 37 °C) were shifted to 10 °C for 4 h, and then returned to 37 °C. Their viability was examined after rewarming for 24 h (see Methods for details). First, we observed that cooling induced cell death in a large fraction of neurons (Fig. 8a). Interestingly, neuronal death was taking place during rewarming (Supplementary Fig. 6a), which is somewhat reminiscent of reperfusion injury, arguing that not the cold per se, but rather the burden associated with restoring cellular functions during rewarming, is the critical challenge facing cold-treated neurons.

A recent study compared cold survival of neurons derived from either hibernating or non-hibernating mammals and reported that "hibernating" neurons survive cold much better than "non-hibernating" ones[52]. Thus, hibernating neurons appear to possess intrinsic mechanisms enhancing cold resistance. Remarkably, treating non-hibernating neurons with certain drugs was shown to compensate, at least partly, for their lower cold resistance[52]. Although house mice, upon starvation, are capable of

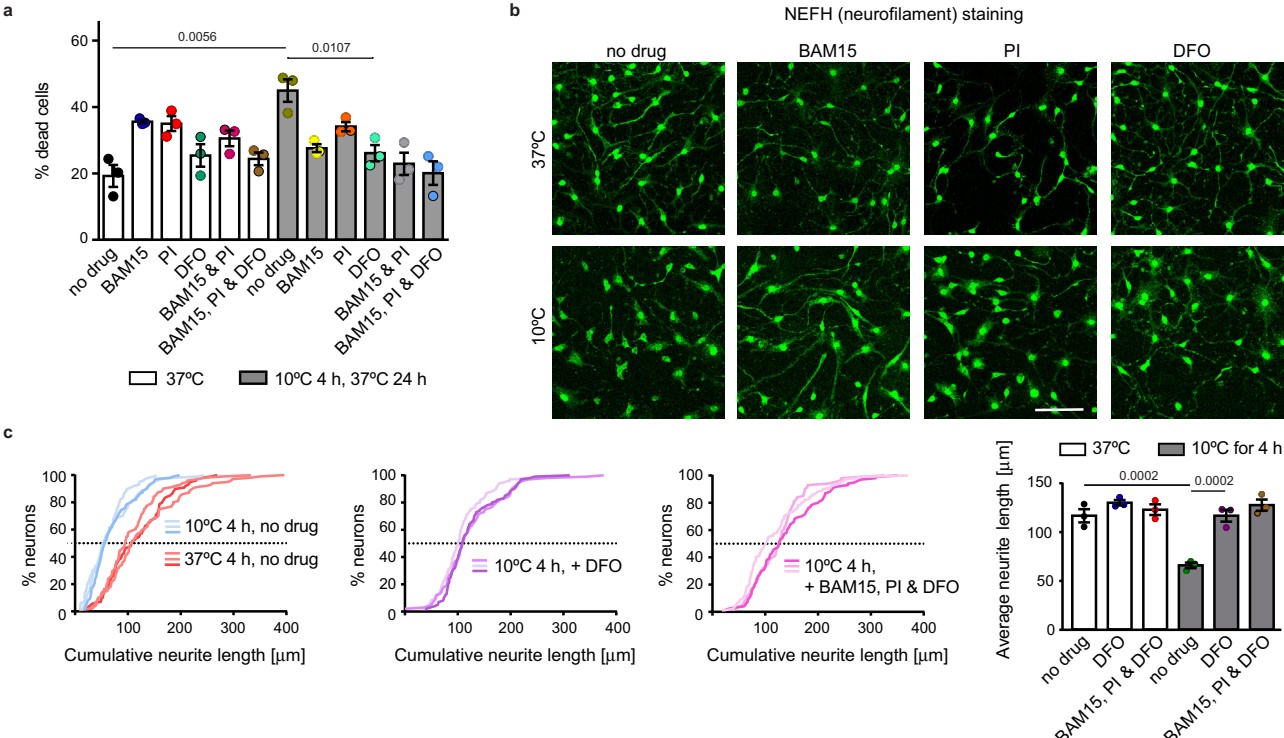

**Fig. 8 | Reducing free iron protects murine neurons from cold-induced degeneration. a** Deferoxamine, similar to BAM15 and protease inhibitors, increases the viability of cold-exposed murine neurons. Cells were incubated continuously at 37 °C or subjected to 10 °C (4 h) with 24 h rewarming (37 °C) in the absence or presence of DFO (100 μM), BAM15 (100 nM), PI (1:500 dilution) or their combinations. Cell viability was examined by staining with propidium iodide. Error bars represent SEM, $n = 3$ independent experiments. 600–800 cells were examined per condition. $p$ values calculated using multiple two-sided $t$-test (left) or two-way ANOVA plus Tukey post hoc test (right). **b** Deferoxamine, similar to BAM15 and protease inhibitors, inhibits cold-induced degeneration of neurites. Shown are representative confocal images of murine neurons incubated at 37 or 10 °C (4 h) with indicated drugs, as in **a**. Cells were stained for NEFH to visualize neurites immediately after cold treatment. Scale bar: 40 μm. **c** Quantification of neurite length, visualized in **b**. The cumulative plots compare the total neurite length of differently treated cells (as indicated), and each curve corresponds to one experimental replicate, $n$. The bar graph (right) compares average neurite length. Error bars represent SEM, $n = 3$ independent experiments. $p$ values calculated using one-way ANOVA plus post hoc Tukey test). Source data are provided as a Source Data file.

daily torpor[53], they can be considered non-hibernators in a classical sense. Correspondingly, we observed that treating murine neurons with either BAM15, a mitochondrial uncoupling drug, or PI, a cocktail of protease inhibitors (drugs previously used by ref. 52), increased their cold survival (Fig. 8a; note that these drugs were beneficial in the cold, while at standard temperature tended to reduce viability). Assuming that, like for nematodes, iron management is crucial for the survival of cold-treated neurons, we treated neurons with the iron-chelating drug deferoxamine (DFO), expected to lower cellular levels of free iron. Consistent with previous studies that, in other cell types, demonstrated cold-protective activity of DFO[22,23,54], we found that DFO treatment protected neurons from cold-related death to the same extent as BAM15, PI, or the combination of these drugs (Fig. 8a).

In contrast to hibernating neurons, the non-hibernating neurons display a striking deterioration of neuronal processes/neurites, which is counteracted by BAM15 and/or PI treatment[52]. By staining neurons against neurofilament protein heavy polypeptide (NEFH; a neuron-specific component of intermediate filaments), we observed that DFO also had a strong stabilizing effect on cold-treated neurites (Fig. 8b, c). This protection appeared to be long-lasting, as the neurites were still evident at 24 h into rewarming (Supplementary Fig. 6b, c).

## Overproduction of FTH1 improves cold survival of mammalian neurons

By lowering the pool of free iron, DFO could indirectly reduce the levels of Fe(II). Nonetheless, to monitor ferrous iron directly, we employed a fluorescent probe, FeRhoNox-1, which specifically detects Fe(II). Strikingly, we observed a strong, though transient, increase of Fe(II) during rewarming (Supplementary Fig. 7a, b), which was reduced by DFO treatment (Fig. 9a). Since ferrous iron catalyzes the formation of ROS, and there are numerous reports on a cold-associated increase in ROS levels[55,56], we also measured ROS levels, using CellROX Green, in cold-treated neurons. We observed a strong increase of ROS during rewarming, at the time coinciding with the Fe(II) peak (Fig. 9b). Importantly, that increase was counteracted by DFO treatment, as expected (Fig. 9b). Moreover, we noticed that the increase of ROS during rewarming was strongly reduced also in cells treated with BAM15 but not PI (Fig. 9b).

If, as expected, decreasing ROS is important for the recovery from cold, treating neurons with antioxidants should provide a similar benefit as DFO. To test that, we selected three therapeutic antioxidants: Edaravone[57–59], NAC[60–62], and TEMPOL[63,64]. Indeed, treating neurons with these drugs strongly enhanced cold survival (Fig. 9c). Finally, we decided to test whether, similar to *ftn-1* overexpression in nematodes, overexpression of its murine counterpart improves cold survival of neurons. The upregulation of ferritin was previously proposed to suppress oxidative damage associated with a cold-induced increase in catalytic iron[65,66]. However, those studies relied on indirect (hemin-induced) upregulation of ferritin and did not discriminate between different forms of ferritin. Thus, we tested specifically the ferroxidase-active murine FTH1 and found that *Fth1*-overexpressing neurons survived cold significantly better than mock-transduced neurons (Fig. 9d). Additionally, cold-exposed neurons overexpressing *Fth1* showed reduced levels of ferrous iron and less ROS, compared with cold-exposed mock-transduced cells (Fig. 9e, f).

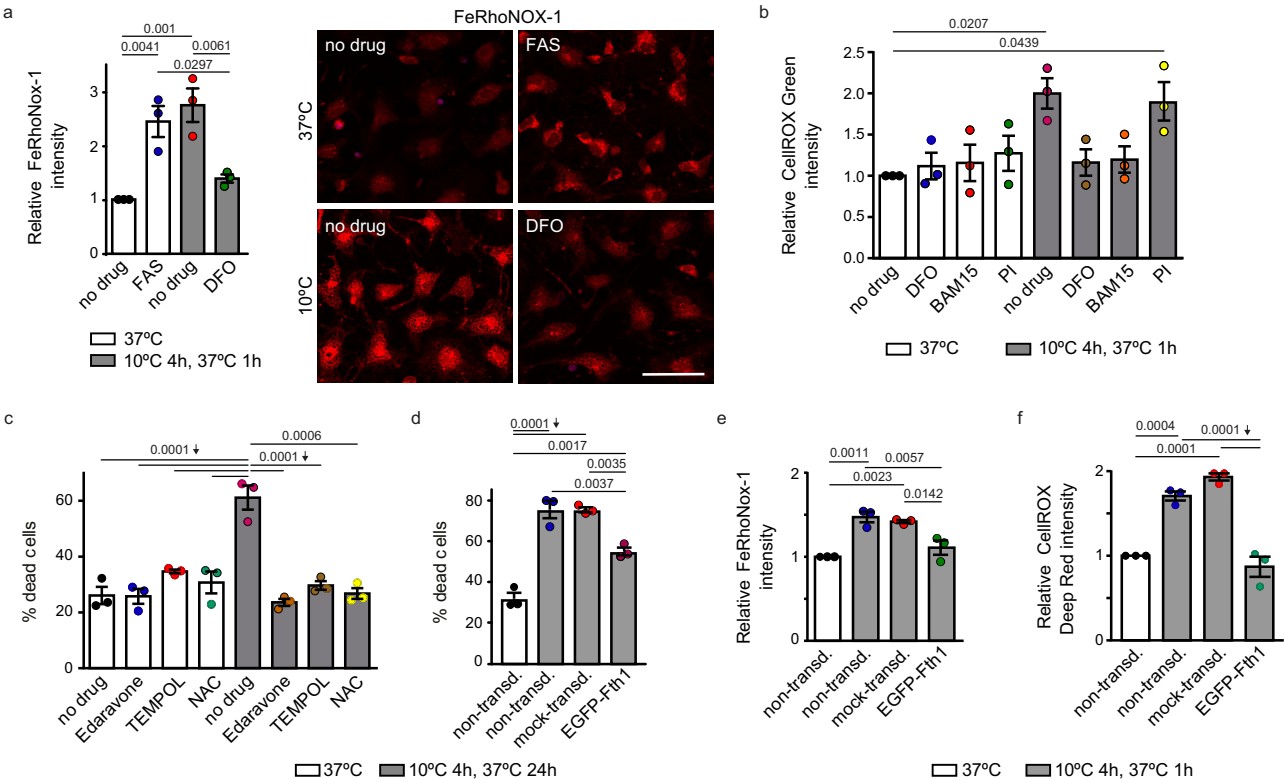

**Fig. 9 | The overexpression of FTH1 promotes cold survival, mimicking drugs targeting the Fe(II)-ROS axis. a** Cold-exposed neurons contain an elevated level of Fe(II). Neurons were incubated at 37 or 10 °C (4 h) with 1 h rewarming (37 °C) in the absence or presence of FAS (100 μM) or DFO (100 μM). Iron(II) was detected using FeRhoNox-1. Left: quantification of FeRhoNox-1 fluorescence, relative to non-treated cells incubated at 37 °C. 150-180 cells were examined per condition. Right: representative images, scale bar: 50 μm. **b** Iron chelation and mitochondrial uncoupling decrease ROS levels in cold-treated neurons. Neurons were incubated as in **a**, in the absence or presence of indicated drugs (concentrations as in Fig. 8a). ROS were detected using CellROX Green, and quantitated relative to non-treated cells at 37 °C. **c** Antioxidants increase the survival of cold-treated neurons. Neurons were incubated at 37 °C or at 10 °C (4 h) with 24 h rewarming at 37 °C, in the absence or presence of Edaravone (50 μM), N-acetylcysteine (NAC, 10 μM) or

TEMPOL (50 μM). Viability was examined by propidium iodide staining. **d** Overexpression of murine FTH1 increases survival of cold-treated neurons. Neurons were transduced with lentivirus carrying either the *EGFP-Fth1* fusion gene or *EGFP* gene (mock-transduced). These and non-transduced controls were treated as in **c**. Viability was examined by propidium iodide staining. **e** FTH1 overexpression decreases the level of iron(II) in cold-treated neurons. Neurons were transduced as in **d**, and treated as in **a**, **b**. Iron(II) was detected using FeRhoNox-1 and quantitated relative to non-transduced cells at 37 °C. **f** FTH1 overexpression decreases ROS in cold-treated neurons. Neurons were treated as in **e**. ROS were detected using CellROX Deep Red and quantitated relative to non-transduced cells at 37 °C. Statistical analysis: Error bars: SEM; replicate number: *n* = 3 (biological in **a**–**e**, technical in **f**); *p* values calculated using one-way ANOVA plus post hoc Tukey test. Source data are provided as a Source Data file.

In summary, cultured neurons appear to respond to hypothermia in a manner remarkably reminiscent of nematodes. Although Fe(II) was only imaged in neurons, both nematodes and neurons appear to display a transient increase in ROS during rewarming. Moreover, induction of ferroxidase-active FTN-1/FTH1 enhances cold survival in both models. Presumably, this reflects the capacity of both orthologous proteins for iron detoxification and, consequently, for reducing ROS. Importantly, targeting the iron-ROS axis with drugs enhances neuronal cold resistance, suggesting that these and related drugs might prove beneficial in treating hypothermia-associated neurological dysfunctions.

## Discussion
In this study, using as the starting point a mutation that increases cold resistance, we described a cold survival-promoting function of FTN-1/ferritin. We found that the induction of *ftn-1* in cold-treated *ets-4(-)* mutant nematodes depends on two TFs, DAF-16 and PQM-1. Although these TFs were previously shown to promote *ftn-1* expression individually[32,37], our results suggest that, under certain conditions, they can function together. Consistent with this idea, PQM-1 and DAF-16 can associate with the same region of the *ftn-1* promoter. This region does not contain any obvious DAF-16-binding element (DBE) but includes a DAF-16-associated element (DAE) (Supplementary Fig. 3b, c), predicted to

bind PQM-1, which we showed is required for the *ftn-1* induction in the cold (Fig. 5c). Thus, assuming that these TFs associate with the same promoter region in the cold, DAF-16 could be recruited to the promoter (directly or indirectly) by PQM-1. Additional TFs, like ELT-2 and HIF-1, known to regulate *ftn-1* expression, could also play a role in the cold[37,67,68]. This is particularly relevant for the GATA-binding factor ELT-2, as the DAE located within the region of the *ftn-1* promoter associating with PQM-1 and DAF-16 contains a GATA motif that, at least in vitro, binds ELT-2[67]. That said, ELT-2 is a general regulator of intestine-specific gene expression[69,70], which, at standard growth conditions, is apparently insufficient for a robust *ftn-1* expression. Thus, one possibility is that, in cold-treated *ets-4(-)* mutants, PQM-1 could be recruited to the *ftn-1* DAE instead of or in addition to ELT-2.

Once overexpressed, FTN-1/FTH1 confers improved cold resistance on both nematodes and cultured cells. Since neither the loss nor overexpression of FTN-1 appears to impact the total levels of stored iron (also in the cold), we suggest that, rather than through iron sequestration, FTN-1 promotes cold survival through iron detoxification, i.e., the conversion of ferrous iron into its ferric form (see model in Fig. 10). This is not meant to say that FTN-1 does not bind iron. Indeed, in agreement with previous studies on human ferritins[44,46], our results suggest that both wild-type and FeOx-dead FTN-1 can bind and

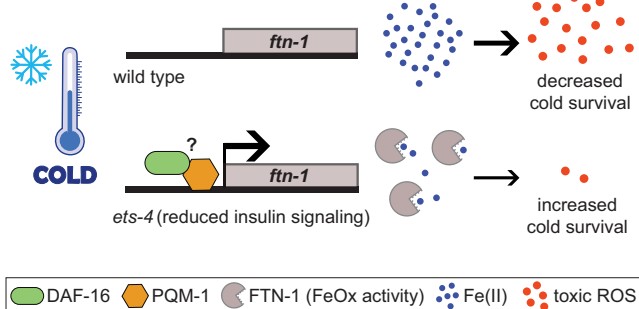

**Fig. 10 | A model for FTN-1/FTH1-dependent cold resistance.** FTN-1, which at standard growth conditions is weakly expressed, is upregulated in cold-treated *ets-4* mutants. This induction requires a combined action of DAF-16 and PQM-1 TFs, where DAF-16 could be recruited to *ftn-1* promotor indirectly, possibly through the interaction with PQM-1. Once expressed, FTN-1, thanks to its ferroxidase (FeOx) activity, converts toxic ferrous iron to its ferric form. By doing so, FTN-1 helps to reduce the levels of cytotoxic reactive oxygen species (ROS), whose production is catalyzed by ferrous iron, thus facilitating cold survival. Overexpression of the mammalian counterpart of FTN-1, FTH1, has a similar effect on cold-exposed neurons (not shown in the model), suggesting that ferroxidase-active ferritin functions as an endogenous antioxidant and, by doing so, promotes cold survival.

store ferric iron. However, when ferrous ions are the source of iron, the FeOX activity speeds up the transition from free to stored iron. Thus, we speculate that FTN-1, through its FeOx activity, facilitates the conversion of ferrous into ferric iron, which then remains, at least temporally, stored within FTN-1, thus efficiently removing the "toxic" ferrous iron from the surrounding. This model does not explain why the inducible FTN-1 is used in addition to the constitutively expressed FTN-2, which is also ferroxidase-active. Among possible scenarios, the de novo-produced ferritin could be more effective than the pre-existing, iron-charged ferritin in detoxifying iron, or FTN-1 may play a role in specific cells or subcellular locations. Crucially, the antioxidant defense is a hallmark of hibernation, being particularly critical during the entry to and exit from hibernation, when oxygen-sensitive tissues, like the brain, are particularly vulnerable to ischemia/reperfusion injury[4]. Intriguingly, the elevated expression of *FTH1* has been recognized as a distinctive feature of cold adaptation in hibernating primates during both daily torpor and seasonal hibernation[71,72]. Thus, the induction of ferroxidase-active ferritin, which in our studies we turned on by genetic manipulations, could be used in nature by some animals as an endogenous antioxidant, boosting organismal cold resistance.

How exactly and why cold triggers the accumulation of toxic iron remains to be fully understood. In mammalian epithelial cells, a sizeable fraction of cold-induced free iron was proposed to originate from the microsomal cytochrome P-450 enzymes, which require iron-containing heme as a cofactor[22]. It was also suggested that the cytosolic free iron causes mitochondrial permeabilization, resulting in apoptosis[54]. However, as shown by ref. 52 and confirmed here, treating neurons with BAM15, a mitochondrial uncoupler, suppresses both ROS and cold-induced death. Assuming that the same treatment reduces iron(II) levels, it may suggest the mitochondrial origin of iron toxicity, agreeing with the general view that most cellular ROS originate from mitochondria[73]. Irrespective of its origin, once produced, iron(II) catalyzes the formation of ROS that damage diverse cellular components (like lipids, proteins, and nucleic acids), which underlies various degenerative conditions[74,75]. Thus, the challenges associated with cooling are, to some extent, similar to those facing the brain in other pathologies. Indeed, the antioxidants employed here have been used to improve outcomes of acute ischemic stroke (Edaravone)[59], hypoxic-ischemic encephalopathy (NAC)[62], and iron-induced cerebral ischemic injury (TEMPOL)[63]. Therefore, new interventions targeting the iron-ROS axis, i.e., preventing the generation of ferrous iron or dealing with

ROS and their consequences, could benefit not only hypothermia patients but be potentially useful in treating other pathologies, like stroke or neurodegenerative disorders.

## Methods

### *C. elegans* handling and genetic manipulation
Animals were grown at 20 °C on standard NGM plates and fed with the OP50 *E. coli* bacteria[76]. All strains used in this study are listed in Supplementary Table 1. The CRISPR/Cas9 genome editing was used by SunyBiotech to generate the *ftn-1* ferroxidase-dead mutant (allele *syb2550*) to tag *daf-16* and *pqm-1* (*syb707* and *syb432*, respectively), and to create a deletion (ΔTGATAAG) in the *ftn-1* promoter (*syb4641*). The tagging was achieved through the C-terminal, in-frame insertion of GFP-FLAG (*daf-16*) or mCHERRY-MYC (*pqm-1*). The FTN-1 over-expressing strains (*sybSi67* and *sybSi72*) were generated (by SunyBiotech) using the MosSCI method, utilizing the insertion locus ttTi5605. The *ftn-1* OE constructs were generated using the MultiSite Gateway Technology and contain circa 2 kb of *dpy-30*, or 1.4 kb of *vit-5* promoter, the genomic *ftn-1* DNA, and 0.7 kb of the *unc-54* 3′UTR. The *sod-5::GFP* strain, GA411, was kindly provided by David Gems.

For RNAi experiments, 1 mM IPTG was added to an overnight culture of RNAi bacteria. About 300 µl of bacterial suspension was plated onto agar plates containing 100 µl/ml of Carbenicillin and 1 mM IPTG. The L4440 (empty) vector was used as a negative RNAi control. Animals were typically placed on RNAi plates as L1 larvae and then were grown to day 1 adulthood at 20 °C, at which time point they were cold-adapted and scored as described. The RNAi clones used in this study came from either Ahringer or Vidal libraries.

### The assay for *C. elegans* cold survival
Unless stated otherwise, all cold survival experiments were performed in the following way: prior to cold adaptation, animals were grown at 20 °C for two generations on OP50. They were then synchronized by bleaching, and L1 larvae were grown until day 1 of adulthood at 20 °C. On day 1 of adulthood, they were cold-adapted at 10 °C for 2 h, and then shifted to 4 °C. Animals were sampled at indicated intervals, and their survival was scored after 24 h recovery at 20 °C. All experiments were performed in three to four independent biological replicates, defined as experiments performed on different days, using separate batches of nematodes. Several hundred animals (200–800) were used for scoring viability at each time point of an individual survival curve. Pairwise Wilcoxon signed-rank test, in R, was used for statistical comparison of survival curves between strains. Original counting data and statistical results are included in Supplementary Data 2.

### Poly-A mRNA sequencing
Around 1000 of 1-day-old adult *C. elegans* were collected after 24 h cold exposure. All steps up to Trizol collection were performed at 4 °C. Animals were washed two times in M9 buffer and snap-frozen in Trizol. Samples were then lysed by freeze/thaw cycles, and RNA extraction proceeded as described before[77]. Genomic DNA was removed using RNeasy Plus Mini Kit (Qiagen). The quality of RNA was monitored by a Bioanalyzer RNA Nano chip (Agilent Technologies). The library was prepared using the TruSeq Library Preparation Kit (Illumina). Poly-A mRNA was sequenced using a Hiseq 50-cycle single-end reads protocol on a HiSeq 2500 device (Illumina). Raw RNA sequence data were deposited at GEO (see data availability statement).

### Genomic data analysis
FASTQC[78] was used to check the quality of the raw sequence data. The reads were mapped to the *C. elegans* genome (Ensembl WBcel235) using STAR[79], with default parameters except: outFilterMismatchNmax 3, outFilterMultimapNmax 1, alignIntronMax 15000,

outFilterScoreMinOverLread 0.33, outFilterMatchNminOverLread 0.33. Count matrices were generated for the number of reads over-lapping with the exons of protein-coding genes using summar-izeOverlaps from GenomicFeatures[80]. Gene expression levels (exonic) from RNA-seq data were quantified as described previously[81]. After normalization for library size, log2 expression levels were calculated after adding a pseudocount of 8 ($y = \log2[x + 8]$). Genes with twofold changes in both replicates were considered significantly differentially expressed. The ChIP bigWig files for PQM-1 and DAF-16 were obtained from ENCODE project[34]. EnrichedHeatmap[82] a R/Bioconductor pack-age was used to generate the integrative heat map.

## RT-qPCR

Around 1000 of 1-day-old adult *C. elegans* were collected at 20 °C prior to cold adaptation, or at 1 day/3 days at 4 °C after adaptation, washed three times in M9 buffer at the respective temperature, and flash-frozen in Trizol. RNA was isolated as above. About 300 or 1000 ng of RNA was used to prepare cDNA with the QuantiTect Reverse Transcription kit (Qiagen) or High-Capacity cDNA Reverse Transcription Kit (Applied Biosystems). cDNA was diluted 1:10 or 1:5 and 5 μl or 2 μl was used with the Light Cycler Syber Green master mix (Roche), or AMPLIFYME SG Universal Mix (Blirt), and Ct values were calculated using Light Cycler 480 (Roche). *act-1* (actin) mRNA was used as the reference gene. Statistical analysis on all experi-ments was performed using the GraphPad/ Prism 8. The statistical method used to calculate *p* value is indicated in figure legends. The following primers were used:

*act-1* FW: CTATGTTCCAGCCATCCTTCTTGG or GTTGCCCA-GAGGCTATGTTC, *act-1* RV: TGATCTTGATCTTCATGGTTGATGG or CAAGAGCGGTGATTCCTTC; *ets-4* FW: CTGAGAACCCGAATCATCCA, *ets-4* RV: TCATTCATGTCTTGACTGCTCC; *ftn-1* FW: CGGCCGTCAA-TAAACAGATTAACG, *ftn-1* RV: CACGCTCCTCATCCGATTGC; *daf-16* FW: AAAGAGCTCGTGGTGGGTTA, *daf-16* RV: TTCGAGTTGAGCTTTG-TAGTCG; *pqm-1* FW: GTGCATCCACAGTAAACCTAATG, *pqm-1* RV: ATTGCAGGGTTCAGATGGAG; *ftn-2* FW: GAGCAGGTCAAATCTAT-CAACG, *ftn-2* RV: TCGAAGACGTACTCTCCAACTC; *sod-5* FW: ATTGC-CAATGCCGTTCTTCC, *sod-5* RV: AGCCAAACAGTTCCGAAGAC.

## Fluorescent imaging of *C. elegans* intestinal nuclei

One-day-old adult *C. elegans* were anesthetized in 20 mM levamisole and placed on 2% agar pads. DAF-16::GFP:::FLAG and PQM-1::mCHER-RY::MYC were imaged on a spinning disc confocal microscope: Zeiss AxioImager equipped with a Yokogawa CSU-W1 scan-head, two PCO Edge cameras, a Plan-Apochromat 40x/1.3 oil objective and two 488 and 561 nm laser lines. Laser intensities and exposure times were kept constant for all samples, camera binning was set to 2. Mean fluores-cence intensity in intestinal cell nuclei (three per nematode) was quantified manually with FIJI/ImageJ[83]. The mean fluorescence inten-sities of each nucleus were averaged and represent one data point for each animal. About 10–15 animals were scored per genotype and bio-logical replicate, in total, around 40 animals per condition. Statistical analysis was performed using the GraphPad/ Prism 8. Two-tailed, unpaired *t*-test was performed to calculate the *p* value between conditions.

## Fluorescent imaging of SOD-5::GFP

One-day-old adult *C. elegans* were anesthetized in 10 mM levami-sole and placed on 2% agar pads. The GFP fluorescence was imaged on Axio Imager.Z2 (Carl Zeiss) equipped with Axiocam 506 mono digital camera (Carl Zeiss), and a Plan-Apochromat 63x/1.40 Oil DIC M27 objective. Images, acquired with the same camera set-tings were processed with ZEN 2.5 (blue edition) microscope software in an identical manner and imported into Adobe Illus-trator. About 6–16 animals were imaged per time point and bio-logical replicate.

## Lifespan assay

The lifespan assay was performed in the following way: nematodes were grown at 20 °C prior to cold adaptation for two generations on *E.coli* OP50. Afterward, nematodes were synchronized by bleaching, and L1 larvae were grown until reaching the young adult stage at 20 °C. Then, the animals were cold-adapted at 10 °C for 2 h and shifted to 4 °C for 5 days. Next, animals were transferred back to the standard culti-vation condition (20 °C) and counted for lifespan. The day when nematodes were shifted to 20 °C is counted as day 0. Animals were scored for viability every second day and transferred into fresh plates until animals stopped laying eggs. The lifespan assay calculation was made by dividing the number of alive nematodes from a certain time point by the number of animals from day 0. The scoring continued until all animals were dead. Animals were counted as dead when they did not respond to touch during picking. Pairwise Wilcoxon signed-rank test, in R, was used for statistical comparisons of lifespans between strains. Original counting data and statistical results are included in Supplementary Data 2.

## Cold survival of *C. elegans* in the presence of NAC

The experiment was designed based on the protocols from the fol-lowing papers[48,84,85]. Freshly made 60 mm 2% NGM plates were admi-nistered with *N*-acetylcysteine (NAC, Sigma-Aldrich) solution (fresh, dissolved in sterile $H_2O$, filter-sterilized 100 mM NAC stock solution). The added amount allowed us to obtain a final concentration of 5 mM NAC in agar. The plates were left overnight to dry. The next day, plates were seeded with a fresh OP50 bacteria culture and left to dry over-night. Bleach-synchronized L1-stage wt nematodes were placed on plates and kept at 20 °C to reach the 1-day-old adult stage. Next, ani-mals were transferred to cold and scored for cold survival as described above. The statistics was calculated as for the assay for *C. elegans* cold survival.

## Oil red O staining and analysis

Oil red O staining was performed as published[86]. In brief, 0.5 g of Oil Red O powder was mixed in 100 ml isopropanol for 24 h, protected from direct light. This solution was diluted in water to 60%, stirred O/N, and sterile-filtered (0.22 μm filter). Around 200–300 1-day-old nema-todes were collected in 1 ml of M9 buffer and washed once with M9. Animals were fixed in 75% isopropanol for 15 min with gentle inversions every 3–4 min. 1 ml of filtered 60% ORO was added to the animals after the removal of isopropanol. Staining was performed for 3–6 h on a shaker with maximum speed and protected from light. Stained animals were placed on 2% agar pads and imaged. Imaging and image analysis was performed as described before[17]. Briefly, animals at 20 °C were imaged using a wide-field microscope Z1 (Carl Zeiss) using a 10x objective and a color camera AxioCam MRc (Carl Zeiss). RGB images were first corrected for shading in Zen Blue software (Carl Zeiss). Afterward, images were analyzed using Fiji/ImageJ software suite[83], stitched with the Grid/Collection stitching plug-in[83], and corrected for white balance. After conversion from RGB to HSB color space, red pixels were selected by color thresholding. A binary mask was created with the Saturation channel and applied to the thresholded images. After conversion to 32-bit, zero pixel values were replaced by NaN. The mean intensity of all remaining pixels was used to represent the amount of red staining in the animals. Animals at 4 °C were imaged using Nikon SMZ25 with a DeltaPix color camera and 60x zoom. The raw image was converted from RGB to HSB color space, the back-ground was subtracted, and then the red signal selection was per-formed through the red pixel threshold for the Hue channel between 0 and 7. The next step was to create a binary mask with the saturation channel and use it for the thresholded image. The image was converted to 32-bit and zero pixel values were replaced by NaN. For further analysis, the integrated density of all remaining pixels was used as the signal coming from the stained animal by Oil red O. In the next step,

the area of the animal was calculated. The raw image was converted to 8-bits and background subtraction was made. To measure the surface of the animal, a threshold of 250 was set, and the area of the particles larger than 3000 pixels was measured. The signal from red pixels was compared to the animal area. About 10–30 animals were imaged per genotype and biological replicate. A two-tailed $t$-test was used to calculate the $p$ value between conditions using GraphPad/Prism 8.

### *C. elegans* extract preparation for iron detection

Animals were collected in M9 buffer in a cold room, washed and resuspended in TBS pH 8.0 (2000 nematodes in 50 μl total volume) with proteinase inhibitors (EDTA-free, Roche) in protein LoBind tubes (Eppendorf). Probes were then homogenized at 4 °C in Bioruptor Pico sonicator (Diagenode), using 30 sonication cycles (30 s on/off). After lysis confirmation, by microscopic inspection, probes were centrifuged for 2 h at 21,130×$g$ at 4 °C, and supernatants were transferred to fresh LoBind tubes. Total protein concentration in the soluble fraction was determined by UV absorbance (NanoDrop, Thermo Fisher Scientific). For SEC-ICP-MS analysis, *C. elegans* extracts were diluted to 10 μg/μl concentration and transferred to 2 ml glass vials (ALWSCI technologies) with 50 μl glass inserts with a bottom spring (Supelco) and kept at 4 °C before the analysis.

### Size exclusion chromatography–inductively coupled plasma-mass spectrometry

All experiments were performed using an ICP-MS-2030 Inductively Coupled Plasma-Mass Spectrometer (Shimadzu, Japan), directly coupled to a Prominence LC 20Ai inert system (Shimadzu, Japan)[87]. Time-Resolved Measurement (TRM) software for LC–ICP-MS was used for controlling both ICP and LC analytical systems. The ICP-MS operates at 1000 W, with an 8.0 L min$^{-1}$ argon plasma gas flow, a 0.7 L min$^{-1}$ Ar carrier gas flow, and a 1.0 L min$^{-1}$ Ar auxiliary gas flow. The sampling depth was 5.0 mm, and the chamber temperature was set to 3 °C. Optimized conditions of the collision cell were −90 V of cell gas voltage, 6.5 V of energy filter voltage, and a 9.0 ml min$^{-1}$ cell gas (He) flow rate. The separation was performed using BioSEC-5, 300 A, 5 μm, 4.6 × 300 mm (Agilent, USA) column using 200 mM ammonium nitrate (99.999% trace metal basis, Sigma-Aldrich) pH 8.0 (adjusted by NH$_4$OH, Sigma-Aldrich, Merck group, Poland) as the mobile phase with a flow rate 0.4 ml min$^{-1}$ and run time 10 min. In all measurements, a 10 μl sample loop was used. For iron content quantification in fractions after separation, ferritin (iron) from equine spleen—Type I standard (Sigma-Aldrich) was diluted to 100, 250, 500, 1000, and 2000 μg L$^{-1}$ total metal concentration in mobile phase solution and used to create a standard calibration curve. Iron content in each fraction was normalized to the peak area in the chromatogram. Total iron concentration was determined by direct sample injection (LC–ICP-MS) and quantification based on iron standard solution (Sigma-Aldrich). The standard calibration curve was created using the same iron concentrations as for the SEC-ICP-MS method.

### Measurement of ROS in cold-treated *C. elegans*

Synchronized animals of the indicated genotypes (young adult stage), grown at 20 °C, were cold-adapted at 10 °C for 2 h and then incubated at 4 °C for 24 h. Animals were collected in an M9 medium and ROS formation was detected essentially as previously reported[49]; by incubating animals for 30 min in an M9 medium containing 3 μM fluorescent dye dihydroethidium (DHE). Next, the animals were washed three times in M9 medium and transferred in M9 medium to a 96-well plate suitable for fluorescence measurements. The fluorescence intensity of DHE-stained animals was measured using a Hidex Sense microplate reader ($\lambda_{exc}$ = 490 nm, $\lambda_{em}$ = 590 nm) and normalized to the number of animals in the well (~100/well). Normalized DHE fluorescence was expressed relative to the signal detected in wt animals. The experiment was performed in three technical replicates for each strain.

### Suspension culture of mouse neural stem cells (NSC) using neurospheres

Entire heads of a fetal mouse (C57BL/6; gestation day between E9-11) were isolated and the tissue was fragmented into pieces followed by incubation in Trypsin-EDTA (0.05 %) (Thermo Fisher Scientific, Waltham, USA) for 15 min at 37 °C. The collection of mouse embryonic brain tissues was approved by the Local Ethics Committee at Poznan University of Life Sciences, Poland. The tissue was subsequently transferred to DMEM/10% FCS and triturated by pipetting up and down into single-cell suspension. The cell suspension was transferred on adherent, uncoated tissue culture plates. After 3 h incubation in 5% CO$_2$ at 37 °C, the residual differentiated and non-neural cells readily attached to the bottom of the plate and the floating neural stem cells were collected. The neural cells were transferred onto low-adhesive six-well plates, coated with Poly-HEMA (Poly 2-hydroxyethyl methacrylate; Sigma-Aldrich, St. Louis, USA) using DMEM medium (Thermo Fisher Scientific), supplemented with F-12 (Thermo Fisher Scientific), B-27 (Thermo Fisher Scientific), 100 ng/ml basic fibroblast growth factor (FGF-2, ORF Genetics, Kopavogur, Iceland), 100 ng/ml epidermal growth factor (EGF, ORF Genetics) and 5 μg/ml heparin (Sigma-Aldrich). After 1 day in culture (5% CO$_2$/5% O$_2$), the neural cells formed neurospheres, which were further cultured and passaged weekly using tissue chopper[88].

### Differentiation of NSC to noradrenergic neurons

Neurospheres were differentiated towards noradrenergic neurons using available protocols[89]. The spheres were dissociated by chopping into small cell aggregates and plated onto glass coverslips coated with 0.05 mg/ml poly-D-lysine (Thermo Fisher Scientific) and 3.3 μg/ml laminin (Sigma-Aldrich). Cells were incubated for 5 days in Neurobasal Medium, supplemented with B-27 serum-free supplement, penicillin/streptomycin (all Thermo Fisher Scientific), and neurotrophic factors: 50 ng/ml BDNF, 30 ng/ml GDNF (Peprotech, UK), according to a modified protocol described elsewhere[89].

### Detection of NSC and mature neuronal markers

Neural stem cell and noradrenergic neural cell identity was confirmed by PCR-based detection of neural stem cell (NSC) gene markers: *Sox2*, *Gbx2*, *Cd-81*, *Cdh1*, *S100b*, *Dach1*, *Pax6*, *Olig1*, or neural differentiation markers: *Cspg4*, *DβH*, *Darpp32*, *Nestin*. Moreover the neurospheres were immunostained for neural stem cell markers: Nestin (1: 500; DSHB, Iowa, USA[90]), Foxg-1 (1:100; Abcam, Cambridge, UK), Emx1 (1: 100; Millipore, Burlington, USA) and Emx2 (1:100; Abgent, San Diego, USA) and differentiated neurons for Th (1:100; Abcam), S100b (1:100; Abcam), DβH (1:500; Abcam), Darpp32 (1:50; Abcam).

### Subcloning, lentivirus generation, and transduction for FTH1 overexpression in mouse noradrenergic-like neurons

The mouse *Fth1* gene was introduced into neurons using a lentiviral pLJM1 vector for EGFP fusion. pLJM1-EGFP was a gift from David Sabatini (Addgene plasmid #19319; http://n2t.net/addgene:19319; RRID:Addgene_19319)[91]. It co-expresses EGFP and puromycin resistance and allows for the visualization and selection of transduced cells expressing N-terminally EGFP-tagged FTH1. After RNA isolation from mouse brain tissue, the cDNA template was synthesized using NEBNext Second Strand Synthesis Enzyme Mix for double-stranded cDNA and Phusion® High-Fidelity DNA Polymerase (M0530, NEB). For constitutive expression, the coding sequence of *Fth1* was PCR-amplified (using ProtoScript II Reaction/Enzyme Mix by New England BioLabs, Ipswich, USA).

The following primers with specific Gibson's overhangs were used:

mFth1overexpGibson_F (TCCGGACTCAGATCTCGAGCTCAAGCT TCGATGACCACCGCGTCTCCCTCG), mFth1overexpGibson_R (GAT-GAATACTGCCATTTGTCTCGAGGTCGAGTTAGCTCTCATCACCGTGT

CCC), and cloned into EcoRI-digested lentiviral pLJM1::EGFP vector. Cloning and DNA preparations were done using NEB® Stable Competent *E. coli* (C3040H), according to Gibson Assembly® protocol (NEB).

Lentiviral particles were assembled using a third-generation packaging system. The plasmid pLJM1::EGFP::FTH1 or "empty" pLJM1::EGFP vector, pMDL, pMD2.G, and pRSV/REV were mixed (3:2:1:0.8), and human embryonic kidney 293 cells (HEK 293 T) ($3 \times 10^6$ cells seeded on T75 flask 1 day before) were transfected using a calcium phosphate protocol. Pseudoviral particles in neuronal maintenance medium were collected at 48 and 72 h post-transfection and filtered through a 0.45 μm filter. The aliquots were snap-frozen and stored at −80 °C. Transduction in neurons was done by replacing the culture medium with one enriched in lentiviral particles (pLJM1::EGFP::FTH1 or empty pLJM1::EGFP), collected before and supplemented with polybrene (5 μg/mL). After 24 h incubation, the medium was discarded and a new one with a lentiviral vector was added for a subsequent 24 h. At 48 h after transduction, the medium was replaced with a fresh viral-free medium and at 96 h post-transduction, selection with puromycin was initiated for a further 48 h.

### Cold treatment of noradrenergic-like neurons

The assay was established using two independent humidified airtight cell culture incubators. One water-jacketed type incubator was additionally equipped with a cooler unit (10 °C) and the other incubator was set to 37 °C. Both contained atmosphere control, which was set to 5% $CO_2$/5% $O_2$. If not stated otherwise, differentiated neuronal cultures were placed in a 10 °C-incubator for 4 h and then returned to the 37 °C incubators for an additional 24 h of rewarming. Such a cooling/rewarming paradigm demonstrated a statistically relevant rise of cell death as early as 4 h into cooling, which was used in subsequent assays. To evaluate the neuroprotective effects of compounds, a neuronal culture medium was replaced with a Neurobasal medium without neurotrophic factors supplemented with 100 μM deferoxamine (DFO concentration optimum determined based on dose curve at 10 °C) (Sigma-Aldrich), 100 nM BAM15 (Tocris, Bristol, UK), 1:500 dilution of protease inhibitor (PI) cocktail III (Sigma-Aldrich). All compounds were provided as a single or combined treatment. The effects of antioxidants were tested by supplementation of neuronal maintenance medium with 50 μM Edaravone (Sigma-Aldrich), 50 μM TEMPOL (Sigma-Aldrich), or 10 μM *N*-Acetyl-ʟ-cysteine (NAC; Sigma-Aldrich), following procedures described above. Drugs concentration optimum was determined based on dose curves at 10 °C.

### Propidium iodide staining

After 4 h of cooling at 10 °C and an additional 24 h of rewarming at 37 °C, neurons cultured on glass coverslips were incubated in the presence of 10 μg/ml propidium iodide (PI) (Cayman Chemical, Ann Arbor, USA) diluted in phosphate-buffered saline (PBS), and co-stained with 1 μg/ml Hoechst 33342 (Life Technologies) for 25 min at 37 °C. Cells were then fixed in ice-cold buffered formalin (4%) for 15 min, washed twice in PBS, and placed in a histology mounting medium (Sigma-Aldrich) on a glass slide. The prepared material was imaged using a fluorescence microscope (Leica DMI 4000B, Germany) and LAS X SP8 software. Counting of total cells (blue nuclei) and necrotic cells (red-PI positive and round) was performed on two to three images from three coverslips as replicates. Collected data were statistically analyzed using Prism software (version 6.01 for Windows, La Jolla, CA).

### Detection of reactive oxygen species (ROS) using CellROX probes

ROS were detected using cell-permeable CellROX reagents (Life Technologies, Carlsbad, USA), which become fluorescent only upon their oxidation by ROS. Murine neurospheres were differentiated into neurons in 24-well plates, growing on glass coverslips. Differentiated cells were maintained continuously at 37 °C (group 1) or were exposed to 10 °C for 4 h (group 2), in the absence or presence of DFO (100 μM), BAM15 (100 nM), and protease inhibitor cocktail (PI, 1:500 dilution), similarly as described elsewhere[52]. Subsequently, cold-exposed cells were rewarmed at room temperature for 5 min, and both groups were stained with fluorogenic CellROX Green reagent (5 μM) for 30 min at 37 °C in the dark. The fluorescence of CellRox Green ($\lambda_{exc} = 485$ nm, $\lambda_{em} = 520$ nm) was visualized using a Leica TCS SP5 confocal microscope and LAS X SP8 software. Z-stacks of well-focused confocal images taken at 0.55 μm intervals in the z-axis from four culture areas per each treatment and condition were collected for both groups. Maximal intensity projections of the Z-stack images were used for data analysis. CellROX Green fluorescence intensity is reported relative to control cells incubated continuously at 37 °C in the absence of drugs. Alternatively, differentiated cells were transduced with lentivirus particles, either pLJM1::EGFP::FTH1 (to express EGFP-FTH1) or pLJM1::EGFP ("mock-transduced" to express EGFP only), or left non-transduced. Transduced and non-transduced cells were then exposed to 10 °C for 4 h and then rewarmed at room temperature for 5 min. These rewarmed cells, and the non-transduced cells kept continuously at 37 °C, were stained with fluorogenic CellROX Deep Red reagent (5 μM). The fluorescence of CellRox Deep Red ($\lambda_{exc} = 640$ nm, $\lambda_{em} = 665$ nm) was visualized and analyzed, similarly to CellROX Green. CellROX Deep Red fluorescence intensity is reported relative to non-transduced control cells that were incubated continuously at 37 °C.

### Determination of iron(II) with FeRhoNox-1

Intracellular iron(II) level was measured using the FeRhoNox-1 probe according to the manufacturer's protocol. Cells were cultured in a glass-bottom dish, and exposed to indicated agents in the cold. Next, cells were rinsed twice with HBSS, then 5 μM FeRhoNox™-1 solution (Goryo Chemical, Inc., Sapporo, Japan) was added and incubated at 37 °C for 1 h, in the dark, and then washed twice with HBSS. To track changes in $Fe^{2+}$ over time, neurons were cooled at 10 °C for 4 h, and the probe signal was recorded at 1, 4, and 8 h of rewarming at 37 °C. To examine iron(II) right after cooling (0 h of rewarming), incubation with the reagent was completed at the end of the 4 h cold exposure. The FeRhoNox-1 signal ($\lambda_{exc} = 543$ nm and $\lambda_{em} = 570$ nm) was visualized using a confocal microscope (Leica TCS SP5, Germany) and LAS X SP8 software. The fluorescence intensity of Z-stacked confocal images of neuronal culture (maximal intensity projection of 7 image z-stacks taken at 0.55 μm intervals in the z-axis) was analyzed using ImageJ.

### Neurite tracing

Cultured neurons with or without DFO or BAM15/PI/DFO were fixed with 4% paraformaldehyde, permeabilized, and washed with 0.1% Triton X-100 in phosphate-buffered saline, and stained by antibody against NEFH (1: 50; DSHB, Iowa, USA, RT97). NEFH⁺ neurite paths were traced with the "Simple Neurite Tracer" plugin[92], ImageJ, using Z-stacked confocal images of neuronal culture (maximal intensity projection of seven image z-stacks taken at 0.55 μm intervals in the z-axis). Cumulative frequency plots of neurite lengths for each experimental group were built using GraphPad Prism version 6.01 for Windows, GraphPad Software, La Jolla, California, USA.

### FTN gene cloning and mutagenesis

The ORFs encoding FTN-1 and FTN-2 were amplified by PCR from *C. elegans* cDNA and cloned between NcoI and NotI sites into pET28a plasmid (Novagen) by ligation-independent cloning using In-Fusion® HD Cloning Plus kit (Takara). Mutations within ORFs were introduced by site-directed mutagenesis using PCR splicing by overhang extension (PCR SOEing). The cloning and mutagenic primers are listed in

Supplementary Table 2. Sanger sequencing was used to verify all cloned constructs.

## Expression and purification of recombinant tag-less FTN

pET28-derived plasmid, containing ORF encoding *C. elegans* FTN-1 and FTN-2, either WT or E58K/H61G-mutated, was transformed into *E. coli* strain BL21-CodonPlus (DE3)-RIPL (Agilent) and protein expression was induced with 0.1 mM IPTG at 18 °C overnight. Recombinant FTN variants were purified using a modified protocol based on the procedure used for the purification of mammalian ferritins[93]. Bacteria were lysed at 4 °C in Lysis Buffer (10 mM Tris-HCl, pH 7.4, 100 mM NaCl, 1% Triton X-100), supplemented with 2 mM beta-mercaptoethanol, 1x Complete™ (EDTA-free) protease inhibitor cocktail (Roche) and 10 U/ml Benzonase nuclease (Sigma-Aldrich), then sonicated, and clarified by centrifugation (4 °C, 30 min, 48,000×*g*). Cleared lysate was incubated at 70 °C for 10 min, and heat-denatured proteins were removed by centrifugation (4 °C, 10 min, 48,000×*g*). FTN present in the supernatant was precipitated by adding 2 M acetic acid drop-by-drop to lower the pH to ~4.8 and incubated at 4 °C for 30 min. The precipitate was harvested by centrifugation (4 °C, 15 min, 5000×*g*) and then re-dissolved in Resuspension Buffer (50 mM Tris-HCl, pH 7.4, 100 mM NaCl) supplemented with 2 mM beta-mercaptoethanol, and adjusting pH to ~7.4 by addition of 0.1 M NaOH. Any non-dissolved material was removed by centrifugation (4 °C, 10 min, 48,000×*g*). FTN was additionally purified using Hi-Trap™ SP HP cation-exchange column equilibrated in Storage Buffer (10 mM Tris-HCl, pH 7.4, 100 mM NaCl), and FTN was recovered in the flow-through after elution with Storage Buffer. FTN solution was then extensively dialyzed at 4 °C using centrifugal concentrators (Sartorius, Vivaspin 100 kDa cut-off), first against Resuspension Buffer containing 1% thioglycolic acid to remove trace iron, then against Resuspension Buffer to remove thioglycolate, and finally against the Storage Buffer. Concentrated proteins were aliquoted and stored at −20 °C. Proteins were >95% pure as assessed by SDS-PAGE and Coomassie Blue staining. Protein concentration was determined using Pierce BCA Protein Assay Kit.

## Bioinformatics analysis

Sequence alignments were generated using algorithms embedded in the JalView v2.11.2.2 interface (http://www.jalview.org/)[94] or Clustal Omega website (https://www.ebi.ac.uk/Tools/msa/clustalo/)[95].

## Analysis of protein complexes under non-denaturing conditions

Recombinant FTN variants (2 µg) were resolved using 4–16% gradient NativePAGE™ Bis-Tris gel (Thermo Fisher Scientific), according to the manufacturer's instructions. The gel was stained with Coomassie Blue. NativeMark™ Unstained Protein Standard (Thermo Fisher Scientific) was used to evaluate the size of protein complexes.

## Source of Fe²⁺ ions

Ferrous ammonium sulfate (FAS) was used as a source of $Fe^{2+}$ (ferrous) ions in ferroxidase and iron uptake assays. Solutions of FAS (50 mM stocks) were prepared daily fresh by dissolving the compound in MQ-quality $H_2O$ supplemented with 2 mM DTT.

## Ferroxidase activity of recombinant FTN

The ferroxidase activity of recombinant FTN, i.e., the ability to mediate oxidation of $Fe^{2+}$ to $Fe^{3+}$ (ferric) ions, was followed by the formation of yellow-colored $Fe^{3+}$:acetohydroxamic acid complex[96]. Typically, 400 µL mixtures were prepared in Assay Buffer (0.2 M Na-acetate, pH 6.0), containing DTT (2 mM), acetohydroxamic acid (5 mM), and either Storage Buffer or recombinant FTN (0.1 mg mL⁻¹ ~5 µM monomer). Reaction mixtures were allowed to equilibrate at RT for 10 min, and then the reaction was started by adding FAS (0.05 mM) through manual mixing. The kinetics of the formation of the $Fe^{3+}$:acetohydroxamic acid complex was monitored continuously at 420 nm for 1 min, using a Shimadzu UV-1601 spectrophotometer. The initial rate of complex formation was determined from the slope of the linear part of the kinetic curve (initial 10 s) and expressed as a change in absorption at 420 nm per min ($\Delta A_{420nm}$ min⁻¹). Ferroxidase activity of recombinant FTN is reported relative to the activity detected in samples containing only Storage Buffer, resulting from spontaneous oxidation of $Fe^{2+}$ to $Fe^{3+}$. The experiment was done in three biologically independent replicates, defined as average from four technical replicates.

## Iron binding by recombinant FTN

The ability of recombinant FTN to bind iron was followed by the formation of amber-colored ferric oxide ($Fe_2O_3$) hydrate, which remains soluble due to incorporation into the ferritin cage, and can be quantitated by measuring the absorption at 310 nm, and using a molar extinction coefficient $\varepsilon_{310nm} = 2\ 475\ M^{-1}\ cm^{-1}$ [97,98]. Typically, 400 µL mixtures containing recombinant FTN (0.1 mg mL⁻¹ ~5 µM monomer) were prepared in Storage Buffer (pH 7.4), and the reaction was started by adding FAS (0.25 mM) through manual mixing. The kinetics of $Fe_2O_3$ hydrate formation was monitored continuously at 310 nm for 1 min, using a Shimadzu UV-1601 spectrophotometer. The initial rate of product formation was determined from the slope of the linear part of the kinetic curve (initial 10 s) and expressed as a change in absorption at 310 nm per min ($\Delta A_{310nm}$ min⁻¹). Final data were reported as the initial rate of iron binding relative to that detected in samples containing only Storage Buffer. Kinetic experiments were done in 4 biologically independent replicates, defined as average from two technical replicates. Iron uptake was also monitored by the end-point method, where recombinant FTN (~5 µM) was incubated with FAS (0.5 mM) at RT for 1 h and then centrifuged (RT, 5 min, 20,000×*g*) to remove any insoluble non-ferritin polynuclear iron[98]. The amount of soluble, FTN-bound $Fe_2O_3$ hydrate was determined by measuring the absorption of the supernatant at 310 nm, with samples not containing FAS set as blank. Samples without FTN were used as a control for $Fe_2O_3$ hydrate solubilized by the Storage Buffer itself. The amount of iron bound to FTN is reported relative to that detected in samples containing only Storage Buffer. The experiment was done in four biologically independent replicates, defined as average from three technical replicates.

## Mass spectrometry detection of FTNs in *C. elegans* extracts enriched for FTNs

Nematodes, *ets-4(-)* or *ftn-1(FeOx-mut); ets-4(-)* mutants, were grown at 20 °C on 10 cm diameter NGM plates until reaching the 1-day-old adult stage, then cold-adapted at 10 °C for 2 h and incubated at 4 °C for 3 days. Animals from five plates were collected in ice-cold M9 buffer by centrifugation (4 °C, 1 min, 400×*g*) and washed three times in ice-cold M9 buffer. A wet nematode pellet (~500 µL) was snap-frozen in liquid $N_2$ and stored at −80 °C until further use. Lysis Buffer (500 µL), supplemented with 5x Complete™ (EDTA-free) protease inhibitor cocktail, was added to frozen pellets, which were thawed at 20 °C and vortexed. The suspension was subjected to 5–10 cycles of freezing (liquid N2), thawing (20 °C), and vortexing, then sonicated on ice and cleared by centrifugation (4 °C, 5 min, 16,100×*g*). Protein concentration in cleared lysate was determined using BCA Protein Assay Kit (Pierce). Approximately 5 mg of protein in the lysate was used for further processing. Lysates from different strains were adjusted to equal volume (~800 µL) with Lysis Buffer, then incubated at 70 °C for 10 min, and cleared by centrifugation (4 °C, 5 min, 16,100×*g*). Proteins from the supernatant, including FTNs, were acid-precipitated using acetic acid (final 0.2 M), incubated on ice for 30 min, and precipitates were harvested by centrifugation (4 °C, 5 min, 16,100×*g*). Proteins in

the pellet were resolved by SDS-PAGE. The region of Coomassie Blue-stained gel containing FTNs, around 20 kDa marker, was cut out and subjected to overnight in-gel digestion with trypsin (Promega). Peptides resulting from proteolytic digestion were desalted using OMIX-SPE 10 μl tips and analyzed by a liquid chromatography system (NanoElute) coupled to a TimsTOF Pro mass spectrometer (Bruker Daltonics, Bremen, Germany). Proteins and peptides were identified using *C. elegans* protein sequence database from Swiss-Prot/TrEMBL (27,154 entries), using the PEAKS Studio v10.6 program. The MS signals from FTN-1 and FTN-2 unique peptides were integrated and normalized to the total signal of all background *C. elegans* proteins detected in the sample. Normalized level of FTNs detected in *ftn-1(FeOx-mut); ets-4(-)* is reported relative to *ets-4(-)* mutant. FTN-1:FTN-2 ratio is reported for both mutants. The experiment was performed in three biologically independent replicates.

### Reporting summary

Further information on research design is available in the Nature Research Reporting Summary linked to this article.

## Data availability

Source data are provided with this paper. The RNA sequencing data generated during the current study are available in the GEO repository, with accession No. GSE131870. The mass spectrometry proteomics data have been deposited to the ProteomeXchange Consortium via the PRIDE[99] partner repository, with the dataset identifier PXD034794 and 10.6019/PXD034794. The rest of the data supporting the findings presented in this study are available from the corresponding author upon request. Source data are provided with this paper.

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

## Acknowledgements

We are grateful to Susan Gasser and the Gasser lab for supporting T.P. in later stages of her PhD. We thank Sebastien Smallwood and Stephane Thiry for assistance with mRNA sequencing, Laurent Gelman and Steven Bourke for imaging support, Aneta Dyczkowska for assistance with ORO staining, and Weronika Wendlandt-Stanek for FTN-1 modeling. We thank Bernd Thiede (Department of Biosciences, University of Oslo) for help in MS analysis and the Laboratory of Animal Model Organisms (ICHB PAS) for assistance with *C. elegans* reagents. We thank Collin Ewald, Jacek Kolanowski, Gawain McColl, and Göran Nilsson for discussions and comments on the manuscript. The project POIR.04.04.00-00-203 A/16 was carried out within the Team program of the Foundation for Polish Science, co-financed by the European Union under the European Regional Development Fund. RC was also supported by the EMBO Installation Grant No. 3615, the Polish National Science Center grant 2019/34/A/NZ3/00223, and the Research Council of Norway grant FRIMEDBIO-286499. The research leading to these results has received funding from the Norwegian Financial Mechanism 2014–2021 operated by the Polish National Science Center under the project contract nr UMO-2019/34/H/NZ3/00691. K.S.-K. and M.Fi. were supported by the National Science Center grant 2018/31/B/NZ3/03621. Some of the strains were provided by the Caenorhabditis Genetics Center (CGC) funded by the NIH. The Nestin and NEFH antibodies were obtained from the Developmental Studies Hybridoma Bank, created by the NICHD of the NIH, and maintained at the University of Iowa.

## Author contributions

T.P. performed and analyzed most nematode experiments in Figs. 1–5. J.L. and K.S.-K. performed and analyzed experiments in Figs. 8,9, under the guidance of M.Fi., who also provided the neural stem cell-sphere model. A.A.K. and D.S. performed and analyzed nematode experiments shown in Figs. 6,7; M.Fr. oversaw the ICP-MS experiments. Y.G. analyzed the genomic data and performed some nematode experiments. J.M.M. performed in vitro experiments. A.A.D. and W.P. assisted with strain characterization. R.C. conceived and supervised the project. R.C., with other authors, wrote the manuscript.

## Competing interests

The authors declare no competing interests.
