## [Peer Review File · Nature Communications]

Title: Ferritin-mediated iron detoxification promotes hypothermia survival in *Caenorhabditis elegans* and murine neuronsREVIEWER COMMENTS

Reviewer #1 (Remarks to the Author):

This is my assessment of the manuscript “Surviving Hypothermia by Ferritin-Mediated Iron Detoxification”, in which Pekec et al. explain how survival of cold exposure in *C. elegans* can be promoted by the two transcription factors DAF-16 and PQM-1 and their induction of the ferritin FTN-1. They argue that increased FTN-1 levels reduce the abundance of Fe²⁺ that otherwise would generate reactive oxygen species and thereby impair survival. The authors show that this beneficial effect of ferritin promotes also the cold-survival of mammalian neurons and therefore may have clinical use, e.g. for the treatment of hypothermia, etc.

All experiments are well-conducted and the manuscript is well-written, the topic of cold-survival is interesting and relatively underexplored, and the eventual mechanism of ferritin and its effect of lowering Fe²⁺ and ultimately ROS levels is very appealing. Also convincing is the mammalian cell culture work at the end of the manuscript that demonstrates potential relevance of this mechanism for humans. However, what disturbs me is that the ability of iron sequestration to improve cold survival (i.e. by reducing ROS) is not new, but has been described before. See Zieger et al., *Cryobiology* 1990, or Rauen et al., *FASEB Journal* 2000, or Salahudeen et al., *Transplantation* 2001 – just to name a few. Somehow, the authors do not refer to this work. Why?

Also, the experiments leading up to the discovery of ferritins as cold-survival promoters seem then very hypothesis-driven (possibly motivated by this prior uncited work). Especially the identification of FTN-1 as the crucial direct target of DAF-16 and PQM-1 by a combination of transcriptomics and CHIP seems only weakly supported by the data. This part needs additional work.

In summary, the manuscript might be suitable for *Nature Communication*, but only after the authors more clearly acknowledge the fact that a hypothermia-protective capability for iron sequestration, e.g. through ferritin or through drugs, has been suggested before, and after addressing the following points:

Major comments:

1. The manuscript lacks a summarizing model figure. The authors should provide one.
2. Page 12: I don't accept that it was impossible for the authors to measure ROS levels in cold-shocked animals. What about other reagents than DCFDA or what about protein-based sensors like HyPer or Grx1-roGFP2 that are expressed by the animals?
3. Fig. 1C: The authors claim that the cold-shock survival benefit in *ets-4* mutants does not come from changes in fat content. I believe that this figure has been generated from non-cold-shocked animals. Cold-shocked animals should be tested here, as they would be more relevant.
4. Fig. 1E or F: In these experiments, there are still quite some *ets-4* mutant animals that survive, even after 15 days of cold-shock. At least one time, this time-course should be conducted for longer. What happens here eventually? How long of a cold-shock can some of these animals survive? Also, it seems that the variability in cold-shock survival duration between individual animals varies more in *ets-4* mutants. Please comment, why this could be.
5. Fig. 1E/F: The authors claim the effects of DAF-16 and PQM-1 to be non-additive. To me this is not so clear. Is the difference between *pqm-1*; *ets-4* and *daf-16*; *pqm-1*; *ets-4* mutants insignificant? If it is

significant, then the authors should adjust the text to accommodate the possibility of partially independent actions.

6. Fig. 2: Conventional lines expressing fluorescently tagged DAF-16 and PQM-1 were already available in the worm community. 1) Why did the authors not use those lines but go through the effort of generating new lines by CRISPR technology? Certainly, these new lines resemble more physiological protein expression levels, but why make that effort? 2) Since these lines are quite different from the lines used by the major papers on DAF-16 and PQM-1 in the literature, the authors should validate that their transgenes are functional and that their lines behave in identical ways, e.g. when tested for translocation behavior under high and reduced insulin signaling.

E.g. what strikes me is that in Fig. 2B, PQM-1 is not in the nucleus in wt animals at 20C. This is different from Tepper et al, Cell 2013.

7. Fig. 3 and S3: This is arguably the weakest data of the manuscript.

1) For S3B, Cluster1: Does the proper DAF-16-bound element come up significantly or not? The list is truncated already after 10 motifs, while there might still be more significantly enriched ones.

2) Fig. 3B: Why did you not test all 7 candidate genes but only 5? Please explain/comment.

3) While the mRNA-seq seems fine, the ChIP data derives from different conditions. The authors also acknowledge that. I think that to solidify the idea that DAF-16 and PQM-1 directly and jointly bind the promoters of several *ets-4*-mutant-upregulated genes, a proper ChIP-seq for DAF-16 and PQM-1 under conditions that match the mRNA-seqs should be conducted (i.e. in *ets-4* mutants after cold-shock).

Finally, a figure panel of the FTN-1 as well as FTN-2 loci in a genome browser should be shown, displaying the locations of any DAF-16 or PQM-1 binding sequence motifs, the read densities from the new ChIP-seqs, and the peaks called from that ChIP-seq data, illustrating promoter binding by these transcription factors.

4) Fig. 3A/B: Here the authors try to identify which genes co-regulated by DAF-16 and PQM1 are contributing to the reported cold resistance. The candidate of choice is *ftn-1*, even though *ftn-1* RNAi also decreased the viability of WT worms. In Supplement figure 1E it is shown that neither DAF-16 nor PQM-1 are required for the cold survival in WT worms. Please discuss/explain.

5) Fig. 3B: *ftn-1* RNAi impairs cold survival. But what about its effect on lifespan in wild-type and *ets-4* animals? Can the authors exclude that knocking down ferritins simply makes animals sick or shorter-lived?

6) Fig. 3C and D: Would be great if the authors could check protein levels for FTN-1 and 2 and not just mRNA levels.

8. Figure 4 is missing proof that CRISP/Cas9 editing resulted truly in FTN-1 with impaired ferroxidase activity and not in some generally non-functional protein. It needs to be shown that other functions of the protein like iron sequestration were not affected. Do in-vitro assays to address this. Also show that the mutation didn't alter the expression level of the protein.

9. Fig. S4A: Test the effect of FAC concentrations also on lifespan in non-cold-shocked animals. Maybe the increasing amounts of FAC simply make the animals sick/short-lived.

10. Fig. S4B: This result could also be explained by FAC simply being toxic and thereby impairing survival. See my point 9.

11. Figure 5C: See my point 2. The authors should look for a better ROS sensor. I find following *sod-5* expression unsuitable for the purpose of monitoring ROS level changes.

12. Inspired by Fig. 7, the authors should test if administration of antioxidants improves viability of *C. elegans* after cold-shock. This would be expected.
13. Fig. 6C: The drug exposure effect on cells that were not cold-shocked but grown at 37C should be quantified here, too. It needs to be excluded that the drugs promote neurite length increase in general. The same issue applies to Fig. S6C.
14. Figure 7B and C: The effect of these drugs on cells not exposed to cold-shock should also be shown.
15. Figure 7: Add measurements of ROS or Fe²⁺ levels in cells which overexpress Fth1, to show that Fth1 really affects ROS and Fe²⁺.

Minor comments:

1. Page 8, line 2: Suddenly FTN-1 is mentioned, but it has not been introduced before. This should be corrected in the text.
2. Fig. S1C: This panel should be moved to main Figure 1, near panel 1D.
3. In Fig. 4A, red colors are too similar. Please change.
4. Fig S3: Please provide the number of genes in each cluster.

Reviewer #2 (Remarks to the Author):

Pekec and colleagues described in this study how enhanced expression of ferritin can increase cold survival in *C. elegans* and mammalian neurons. They started with the cold resistance phenotype of *ets-4* mutants and identified DAF-16 and PQM-1 as regulators of a key downstream effector gene *ftn-1* in promoting cold survival in *C. elegans*. Several lines of evidence suggest that ferritin encoding *ftn-1* acts by iron detoxification and promotes survival by reducing iron-mediated generation of reactive oxygen species (ROS). They show both genetic and pharmacological means to reduce iron-mediated ROS can also reduce death and degeneration of mammalian neurons under cold conditions. Overall this is a straightforward paper with well-designed and executed experiments showing interesting results with potentially important biomedical implications. Major concerns center on conceptual novelty of key findings and several issues that need to be addressed before being suitable for publication.

1, Roles of *ftn-1* in iron-mediated ROS detoxification, stress resistance and longevity and its regulation by the DAF-16 pathway have been previously reported by several publications (e.g. PMID: 22445852; PMID: 22396654). Cold stress is also widely known to cause cell damage by ROS. In addition, BAM15 and other antioxidants against iron-mediated ROS have also been previously shown to increase cold survival of mammalian neurons (PMID: 29576452). It is questionable how much novelty it is conceptually or mechanistically to establish a link between ferritin to resistance of cold stress.

2, Cold stress especially under severe cold conditions like 2-4 C used in the paper is known to arrest most *C. elegans* behaviors including locomotion, pumping and feeding, causing essentially a starvation like response. This is also consistent with the known role of DAF-16 in mediating starvation response and activating *ftn-1*. It would be important to examine if the regulatory axis the author proposed mediates a specific response to cold or a general response to starvation.

3, *ftn-1* has been previously reported to be regulated by several other transcription factors including HIF-1 and ELT-2 (PMID: 18024960; PMID: 22194696), in addition to DAF-16. That literature should have

been at least cited and discussed. Is the phenotype described in this study dependent on HIF-1 and ELT-2?

4, As this study focuses on *ftn-1* regulation and roles in cold resistance, it would be more appropriate to expand analysis of *ftn-1* including key questions of importance on how DAF-16/PQM-1/*ftn-1* might be regulated by cold, specificity of regulation in stress types, and functional consequences on different tissues especially cold stressed neurons in *C. elegans*. Figure 1 and 2 on less important ETS-4 and DAF-16 should be moved to supplementary data.

5, The authors conclude that FTN-1 promotes cold survival via its ferroxidase activity. Did the author test the assumption that enzymatic site mutations affect only ferroxidase activity rather than gene expression, protein abundance, protein-protein interaction or cellular localizations?

Reviewer #3 (Remarks to the Author):

The present study provides evidence that ferritin promotes cold survival in *C. elegans* and in murine neurons. Ferritins are ferroxidases that catalyzes the oxidation of Fe(II) to Fe(III) and sequester iron within the ferritin shell, thereby protecting cells from iron-induced oxidative damage. Using genetic approaches and RNA-seq analysis, the authors show that the transcription factors DAF-16/FoxO and PQM-1, which act in the insulin-IGF-1 signaling pathway, upregulate ferritin during cold stress. *C. elegans* express two ferritins, FTN-1 and FTN-2, both of which contain ferroxidase active sites, but only *ftn-1* is upregulated by the insulin-IGF-1 signaling pathway. Several experiments were performed to determine how FTN-1 promotes cold survival. Mutation of FTN-1 ferroxidase active sites abolished cold survival in *C. elegans*, showing that ferroxidase activity is essential in enhancing cold survival. Of note, the authors showed that enhanced cold survival occurred by iron detoxification and not by iron sequestration within ferritin. Further studies revealed that overexpression of mammalian FTH1, chelation of iron or lowering reactive oxygen species in murine neurons protects against cold stress mediated injury. Overall, the findings in this manuscript are interesting and novel in that they show a role for both DAF-16 and PQM-1 in promoting cold survival via upregulation of *ftn-1* as well as a novel function for FTN-1 in iron detoxification rather than iron sequestration. The extension of the studies to murine neurons show an important function for ferritin in cold survival in murine neurons, which may have therapeutic implications.

Major comments

1. For readability and clarity, some figures and sections should be reorganized. Several sections describe supplemental data that are essential for understanding the experiments, and should be included in the main body of the manuscript. The manuscript is challenging to read because the reader has to go back and forth between supplemental and main figures. Also, the manuscript require considerable editing.

In addition, some sections could be reorganized. For example, the first section of the manuscript "Inhibition of ETS-4 improves *C. elegans* survival in the cold" contains Fig. 1D-F, which show survival curves with insulin signaling mutants; however, insulin signaling is discussed in the second section "The

enhanced cold survival requires both DAF-16 and PQM-1". A suggestion for reorganization of these sections/figures is described below.

The first section containing Figs. 1A-C could also contain Figs. S1A-B. Together, these figures nicely describe cold survival data.

The second section containing Figs. S1C, S1D and Figs. 1D-F could be a new figure (Fig. 2). The data in these figures show that cold survival requires insulin signaling. New Figure 2 would include Fig. 2A (S1C), 2B (S1D), 2C(1D), 2D(1E), 2E(1F).

2. In the section "Identification of a PQM-1 and DAF-16 coregulated gene promoting cold survival", the heat map/cluster analysis and CHIP ENCODE data in Fig. S3A should be in the main body of the paper. It provides a clear visual presentation of cluster analysis, and PQM-1 and DAF-16 binding. The DAF-1 and PQM-1 like motifs in Fig. S3B could be included in Fig. 3 or in the text. The Table showing enriched motifs could remain in the supplement.

3. Include raw RNA-seq datasets as well as a table in the supplement showing mRNAs altered in the different strains.

4. What are the molecular weights of species in Peak 1 (HMW) and Peak 2 (ferritin) in Fig. S4C-D? Is it known whether FTN-1 or FTN-2 form homopolymers or heteropolymers? Also, include a statement in the legend that homogenates for these experiments are from cold-treated animals.

5. As shown in Fig. S4B, ftn-1 mutants show sensitivity to iron and cold. Authors should assess ftn-2 mutant viability under cold and iron stress similar to ftn-1 experiments.

6. Elaborate on the model for FTN-1 "iron detoxification" vs sequestration. What is the fate of the detoxified iron - is it sequestered in FTN-2? SEC-ICP-MS plots in Fig. S4C-D could be shown in Fig.4.

7. Include a model for the insulin-IGF-1 signaling pathway-mediated regulation of ferritin during cold stress. How do PQM-1 and DAF-16 coregulate ftn-1?

8. Cite reference related to ferritin protection from cold injury in endothelial cells. M. A. J. Zieger and Mahes P. Gupta, Hypothermic preconditioning of endothelial cells attenuates cold-induced injury by a ferritin-dependent process. *Free Radical Biology & Medicine* 46 (2009), 680-691.

Other comments

1. Fig. 1D: for consistency label y-axis "Viability"

2. For survival curves, explain "n". Are they independent biological replicates? If so, explain biological replicates.

3. Lines 173-175: add (Fig. 2A, C) at the end of the sentence. From statistics in Fig. 2C, these data show significance (**p), although it is stated, "possibly with minimal increase".

4. Lines 169-171: add references

5. Line 195: sentence is out of context (Fig. 3C)

6. Lines 180-182: add (Fig. S1C)

7. Line 234: add Tepper et al. reference number

8. Lines 234-236: List the seven genes in the text in parentheses

9. Lines 279-281: Change Figs. 4B, S4C, E to (Figs. 4B, S4C, S4E).

10. Fig. S3B: line 67-68: clarify "start -1500 end 1500-p6"

11. Line 508: “reproductively” should be “reproducibly” in Fig. 3 legend
12. Line 540: delete “Comparing”
13. Line 291: add the amino acid changes for ferroxidase point mutations at end of sentence.
14. Line 310-311: insert Fig. 5C
15. Include SOD-5::GFP images (Fig. S5) in Fig. 5.
16. Fig. S5A: modify y-axis to visualize wt levels.
17. Line 324: correct spelling to applicable
18. Lines 344-345: reference is needed “Remarkably ... lower cold resistance”
19. Line 348: define BAM and PI; add Ou reference
20. In Fig. 6A, add the concentration of DFO, BAM15 and PI and incubation time in figure legend
21. Fig. 6C: curves need better definition
22. Lines 418-419: Include references for antioxidant therapy in humans

We were happy that all reviewers found our work interesting and would like to thank them for their efforts. We found the reviews constructive and useful. Below is our point-by-point response to the comments.

Reviewer 1:

All experiments are well-conducted and the manuscript is well-written, the topic of cold-survival is interesting and relatively underexplored, and the eventual mechanism of ferritin and its effect of lowering Fe²⁺ and ultimately ROS levels is very appealing. Also convincing is the mammalian cell culture work at the end of the manuscript that demonstrates potential relevance of this mechanism for humans. However, what disturbs me is that the ability of iron sequestration to improve cold survival (i.e. by reducing ROS) is not new, but has been described before. See Zieger et al., Cryobiology 1990, or Rauen et al., FASEB Journal 2000, or Salahudeen et al., Transplantation 2001 – just to name a few. Somehow, the authors do not refer to this work. Why?

Admittedly, we were too nematode-centric and apologize for this bias. These and additional papers are now cited in various parts of the manuscript: Refs. 22, 23, and 24 (Introduction); 22, 23, 51, 53, 64, and 65 (Results); and 22, 51, and 53 (Discussion). We believe that our findings strengthen and extend those earlier studies and, importantly, demonstrate their importance in intact animals.

Also, the experiments leading up to the discovery of ferritins as cold-survival promoters seem then very hypothesis-driven (possibly motivated by this prior uncited work). Especially the identification of FTN-1 as the crucial direct target of DAF-16 and PQM-1 by a combination of transcriptomics and ChIP seems only weakly supported by the data.

Following the observation that the enhanced cold survival of *ets-4* mutants depends on both DAF-16 and PQM-1, we took a functional genomic approach to identify the responsible genes. The primary screen, i.e., the combination of expression profiling and ChIP, revealed candidate genes, which, like in any other screen, were subsequently validated by additional experiments. Thus, our conclusions are based on several complementary lines of evidence. The genetic experiments, which demonstrated that FTN-1 is required (in *ets-4* mutants) and sufficient (when overexpressed in wild type) for the enhanced cold survival, are supported by both old and new data (see our responses to specific points below). All combined, we believe they strongly support the cold survival-promoting activity of FTN-1.

This part needs additional work. In summary, the manuscript might be suitable for Nature Communication, but only after the authors more clearly acknowledge the fact that a hypothermia-protective capability for iron sequestration, e.g. through ferritin or through drugs, has been suggested before, and after addressing the following points:

We hope that the relevant literature is now acknowledged and cited where appropriate. Our point-by-point responses to the remaining comments follow below.

Major comments:

1. The manuscript lacks a summarizing model figure. The authors should provide one.

We include now a summarizing model in Fig. 10.

2. Page 12: I don't accept that it was impossible for the authors to measure ROS levels in cold-shocked animals. What about other reagents than DCFDA or what about protein-based sensors like HyPer or Grx1-roGFP2 that are expressed by the animals?

Following extensive troubleshooting, we decided to use DHE (Dihydroethidium) in ROS measurements, which was tolerated better by cold-exposed animals (Fig. 1 for Rev. 1). We did observe the expected trend, i.e., less ROS in *ets-4* mutants than wt, and comparable to wt levels of ROS in *ftn-1* or *ftn-1; ets-4* mutants. However, the variation was large and so the *p* val between wt and *ets-4* = 0.0547 (t-test), just above the significance threshold (0.05). If this satisfies the Reviewer, we would include this data as Fig. 7f.

3. Fig. 1C: The authors claim that the cold-shock survival benefit in *ets-4* mutants does not come from changes in fat content. I believe that this figure has been generated from non-cold-shocked animals. Cold-shocked animals should be tested here, as they would be more relevant.

This experiment was to exclude that *est-4* mutants survive cold better than wt because of their higher fat content. This was not the case at 20°C (current Fig. 1e). As requested, we now measured fat also in the cold (current Fig. 1f). We selected day 3 in the cold, when both wt and *ets-4* animals fully recover when rewarmed. As shown in Fig. 1f, the *ets-4* mutants had, in the cold, less fat than wt. Thus, these animals survive cold better *despite* containing less fat.

4. Fig. 1E or F: In these experiments, there are still quite some *ets-4* mutant animals that survive, even after 15 days of cold-shock. At least one time, this time-course should be conducted for longer. What happens here eventually? How long of a cold-shock can some of these animals survive?

A longer cold survival experiment was shown before in Fig. S1c, now in Fig. 2a (moved after a request from another reviewer). There, we monitored *ets-4* mutants for 23 days, at which time point all *ets-4* animals were eventually dead.

Also, it seems that the variability in cold-shock survival duration between individual animals varies more in *ets-4* mutants. Please comment, why this could be.

There is inherent variability in cold-survival experiments, not just in *ets-4* animals but also in other strains. We noticed that crowded incubator has tendency to exacerbate variability, particularly in plates surrounded by others, possibly due to locally reduced oxygen concentration. There may also be some impact of humidity. We take great pains to minimize variability, performing experiments in a similar manner and monitoring temperature throughout the time course. Nevertheless, some variability seems inevitable, which is why the experiments were performed in several biological replicates and were analyzed for significance.

5. Fig. 1E/F: The authors claim the effects of DAF-16 and PQM-1 to be non-additive. To me this is not so clear. Is the difference between *pqm-1; ets-4* and *daf-16; pqm-1; ets-4* mutants insignificant? If it is significant, then the authors should adjust the text to accommodate the possibility of partially independent actions.

The difference between the two strains (now shown in Fig. 2e) is small but significant (p value = 0.01, in Table S2). We adjusted the text accordingly.

6. Fig. 2: Conventional lines expressing fluorescently tagged DAF-16 and PQM-1 were already available in the worm community. 1) Why did the authors not use those lines but go through the effort of generating new lines by CRISPR technology? Certainly, these new lines resemble more physiological protein expression levels, but why make that effort?

This comment refers to current Fig. 3. As pointed out by the Reviewer, tagging of endogenous genes is, in general, more physiological, which for us justified the effort. We hope that these lines will be valuable for the whole community interested in these proteins. To our knowledge, most (or all?) of the pre-existing strains express tagged DAF-16 or PQM-1 from gene arrays, which contain from hundreds to thousands of gene copies. These arrays are epigenetically unstable and, in expression studies, need to be used cautiously due to a massive overexpression.

2) Since these lines are quite different from the lines used by the major papers on DAF-16 and PQM-1 in the literature, the authors should validate that their transgenes are functional and that their lines behave in identical ways, e.g. when tested for translocation behavior under high and reduced insulin signaling. E.g. what strikes me is that in Fig. 2B, PQM-1 is not in the nucleus in wt animals at 20C. This is different from Tepper et al, Cell 2013.

As requested, we tested the behavior of these proteins in our strains. We first tested if our tagged DAF-16 is sensitive to insulin signaling. In agreement with previous studies, we observed the nuclear translocation of the tagged DAF-16 upon *daf-2* RNAi (Fig. 2 for Rev. 1). Also similar to Tepper et al., we observed a much weaker accumulation of nuclear DAF-16::GFP upon *pqm-1* RNAi.

As for PQM-1, we indeed did not see the nuclear signal in wt at 20°C with our strain, which was reported with the strain from Tepper *et al.* The difference could be explained simply by the difference in the *pqm-1* gene copy number (one endogenous copy in our strain versus great many in the Tepper strain, where tagged PQM-1 is expressed from an array). However, more relevant for this study, we examined the Tepper strain in the cold and, similar to our strain, observed very clearly the nuclear enrichment of PQM-1::GFP (Fig. 3 for Rev. 1). Thus, we believe the expression patterns we show reflect the genuine nuclear enrichment of DAF-16 and PQM-1 in the cold.

Fig. 3, for Reviewer 1. Nuclear enrichment of PQM-1::GFP (strain from Tepper *et al.*) in cold-exposed animals. Note the nuclear signal increasing with time spent in the cold. Scale bar 20 μ m.

7. Fig. 3 and S3: This is arguably the weakest data of the manuscript.

1) For S3B, Cluster1: Does the proper DAF-16-bound element come up significantly or not? The list is truncated already after 10 motifs, while there might still be more significantly enriched ones.

This comment refers to current Fig. S3a. The analysis produced about 40 enriched motifs. However, the lesser the ranking the more likelihood of false positives. It is customary to select top scorers, which we did, arbitrarily selecting the top 10. Each motif is labelled after the most highly ranked TF. Similarly, the motif labelled as “EGL-5” (4th from the top) also associates with DAF-16 (ranked 3rd within the “EGL-5”-type motif). So yes, the enrichment is significant and the specific prediction of DAF-16 motif is shown now in Fig. S3a. It can be visually compared with the DBE motif now shown in Fig. S3b.

2) Fig. 3B: Why did you not test all 7 candidate genes but only 5? Please explain/comment.

This comment refers to current Fig. 4c. We have now included in the text the names of all seven candidate genes. We tested 5/7 candidates because, in the existing RNAi collections, there were no bacterial strains targeting the two remaining genes (*cpt-4* and *pals-37*). We thus focused on the positive hit, *ftn-1*, which emerged from the analysis. Nonetheless, prompted by the Reviewer, we constructed bacterial feeding strains targeting the two remaining genes. However, we did not observe any consistent effects on cold survival.

*3) While the mRNA-seq seems fine, the ChIP data derives from different conditions. The authors also acknowledge that. I think that to solidify the idea that DAF-16 and PQM-1 directly and jointly bind the promoters of several *ets-4*-mutant-upregulated genes, a proper ChIP-seq for DAF-16 and PQM-1 under conditions that match the mRNA-seqs should be conducted (i.e. in *ets-4* mutants after cold-shock). Finally, a figure panel of the FTN-1 as well as FTN-2 loci in a genome browser should be shown, displaying the locations of any DAF-16 or PQM-1 binding sequence motifs, the read densities from the new ChIP-seqs, and the peaks called from that ChIP-seq data, illustrating promoter binding by these transcription factors.*

We agree with the Reviewer that the ChIP data from the cold would be more direct. However, our goal was to identify, in a screen-like fashion, candidate genes responsible for the increased cold survival, rather than comprehensively determine the DAF-16 and PQM-1 chromatin binding landscape in the cold. The candidate that emerged from this analysis, *ftn-1*, was evaluated in a number ways and

the results are consistent with its co-regulation by DAF-16 and PQM-1. Nevertheless, we did try to redo the ChIP in the cold. To do this with more physiological protein levels, we used our strains, i.e., expressing endogenously tagged DAF-16 and PQM-1. However, the levels of endogenous DAF-16 are too low to allow the detection by western blotting and PQM-1, following crosslinking (part of the ChIP protocol) became insoluble, despite various attempts to solubilize it. It is possible that a lab more experienced with ChIP would have been more successful. However, after attempting this for several months, we took a different approach explained below.

As requested, we show now in Fig. S3c the DAF-16 and PQM-1 ChIP peaks over *ftn-1* and *-2* genes (it is still from the published data at normal temperature). Consistent with our qRT-PCR data (suggesting the regulation of *ftn-1*, but not *-2*, by DAF-16 and PQM-1), we observed ChIP peaks over the *ftn-1* (but not *-2*) promoter. Interestingly, there is one region of the *ftn-1* promoter that recruits both DAF-16 and PQM-1, even though this region only contains a putative PQM-1 binding sequence (TGATAAG) but not a DAF-16 binding sequence (possibly implying an indirect recruitment of DAF-16). We thus created (by CRISPR/Cas9 editing) a new strain carrying a deletion of the TGATAAG sequence and found that *ftn-1* was no longer induced in that strain in the cold (new Fig. 5c). Following a recommendation of another reviewer, we now included a paragraph on transcriptional regulation of *ftn-1* and discuss this new data in the light of previous findings.

4) Fig. 3A/B: Here the authors try to identify which genes co-regulated by DAF-16 and PQM1 are contributing to the reported cold resistance. The candidate of choice is ftn-1, even though ftn-1 RNAi also decreased the viability of WT worms. In Supplement figure 1E it is shown that neither DAF-16 nor PQM-1 are required for the cold survival in WT worms. Please discuss/explain.

The comment refers to current Fig. 4c and Supplementary Fig. 1. We showed that DAF-16 and PQM-1 induce the expression of *ftn-1* in *ets-4* mutants, but not wild type (Fig. 5a), which is why *daf-16* and *pqm-1* mutants (single and double) survive cold like wt (Supplementary Fig. 1). As indicated (now Fig. 4c), the RNAi clone labelled as "*ftn-1, ftn-2*" targets, due to sequence similarity, both *ftn-1* and *-2*. So, in wild type, it is the depletion of both *ftn-1* and *ftn-2* that leads to reduced cold resistance. Our interpretation of these observations is that the non-inducible FTN-2 contributes to the baseline cold resistance in wt, while FTN-1, induced in *ets-4* mutants by DAF-16 and PQM-1, provides an additional resistance.

5) Fig. 3B: ftn-1 RNAi impairs cold survival. But what about its effect on lifespan in wild-type and ets-4 animals? Can the authors exclude that knocking down ferritins simply makes animals sick or shorter-lived?

The comments refer to current Fig. 4c. Answering the latter, we observed that both the *ftn-1* single and *ets-4; ftn-1* double mutants survived cold similar to wild type (Fig. 5d), so the loss of *ftn-1* does not appear to make animals obviously sick. As for the effect on lifespan, neither depletion nor overexpression of *ftn-1* appears to have a consistent impact on the lifespans of animals grown under standard conditions. Valentini et al. (Mech. Ageing and Dev., 2012) concluded that "The effects of loss of *ftn-1* on lifespan (...) are hard to interpret, since depending on genetic background, and mode of abrogation, reduced *ftn-1* expression either increases, decreases or has no effect." Also Kim et al. (JMB, 2004) showed that *ftn-1* depletion had no or only slight effect on the lifespan under normal growth conditions.

We additionally compared the lifespans of wt and *ftn-1*-overexpressing strains, following their cold exposure and recovery. We observed that the *ftn-1* overexpressing animals survived only slightly longer than wt, and this effect was limited to aging animals (after 22 days of incubation; see Supplementary Fig. 5d). Thus, this and previous observations suggest that FTN-1 appears to be more beneficial for cold survival than longevity.

6) Fig. 3C and D: Would be great if the authors could check protein levels for FTN-1 and 2 and not just mRNA levels.

The comment refers to current Fig. 5a and b. To do that, we attempted to create strains expressing either N- or C- terminally tagged endogenous FTN-1. However, the fusion proteins turned out to be inactive, at least in the context of cold survival. We are aware of several papers describing strains expressing GFP-tagged FTN-1. Based on our experience, tagging FTN-1 (even with a small tag like MYC) inactivates the protein, possibly interfering with the formation of ferritin cages. We nonetheless tried to obtain published strains but obtained just one, in which there was no GFP expression. We had high hopes for a recently published antibody (Ma et al., PLOS Gene, 2021) but, at least in our hands, this antibody is only able to detect a highly concentrated recombinant FTN-1 but not the endogenous protein in worm extracts.

8. Figure 4 is missing proof that CRISP/Cas9 editing resulted truly in FTN-1 with impaired ferroxidase activity and not in some generally non-functional protein. It needs to be shown that other functions of the protein like iron sequestration were not affected. Do in-vitro assays to address this. Also show that the mutation didn't alter the expression level of the protein.

The comment refers to current Fig. 6. To address these points (also brought up by another reviewer), we produced recombinant FTN-1 and -2, either wt or carrying point mutations expected to abolish the ferroxidase activity (FeOx-mut; these are the same mutations that we previously introduced in the endogenous *ftn-1* gene). We noticed that both wt and mutated proteins were expressed at similar levels, and each protein was able to form high molecular weight complexes, probably corresponding to the ferritin cages (Supplementary Fig. 4d). We followed previously published procedures to examine functions of these proteins and observed that, as predicted, both FTN-1 and -2 have the FeOx activity and, expectedly, this activity was abolished in the FeOx-mut proteins (Fig. 6d). Similar to previous studies, we observed that the FeOx-dead proteins bound iron, although with reduced kinetics (Fig. 6e and f). Finally, since we were unable to use antibodies or create functional tagged *ftn-1*, we compared the levels of wt and FeOX-dead FTN-1 (untagged), in *ets-4* animals at day 3 in the cold, by semi-quantitative mass-spectrometry, and found no obvious differences in their levels (Supplementary Fig. 4e). Thus, while the mutations abolish the FeOx activity of FTN-1 (and -2), they appear to do so without major distortions to the proteins.

9. Fig. S4A: Test the effect of FAC concentrations also on lifespan in non-cold-shocked animals. Maybe the increasing amounts of FAC simply make the animals sick/short-lived.

10. Fig. S4B: This result could also be explained by FAC simply being toxic and thereby impairing survival. See my point 9.

According to Valentini et al (Mech Ageing Dev, 2012), adding 9 mM did not but 15 mM FAC did shorten the lifespan. We previously observed that, in wt, 5 mM FAC (so below the concentration that acc. to Valentini et al. affects the lifespan) decreased cold survival, arguing that the toxicity of iron in the cold is not related to the decreased lifespan seen at higher FAC concentrations.

Nonetheless, the comments made us re-think the usefulness of the FAC experiments. They were performed at an early stage, when the goal was to test whether iron may play a role in cold survival. Subsequent experiments showed that to be the case but also demonstrated that FTN-1 promotes cold survival without altering the levels of total body iron, which presumably increase upon supplementation with FAC. Thus, after careful consideration, we decided to remove these experiments from the revised manuscript. We hope the Reviewer will agree.

*11. Figure 5C: See my point 2. The authors should look for a better ROS sensor. I find following *sod-5* expression unsuitable for the purpose of monitoring ROS level changes.*

Responded in point 2.

12. Inspired by Fig. 7, the authors should test if administration of antioxidants improves viability of C. elegans after cold-shock. This would be expected.

We noticed several studies that used NAC on worms. As we already used it on neurons, we decided to test this antioxidant. In previous studies, since antioxidants are unstable, nematodes were transferred onto fresh NAC plates every few days (for example Savion et al., PLOS One, 2018 and Desjardins et al., Ageing Cell, 2016). However, cold-treated animals are fragile and die when transferred to new plates. Thus, we tested cold survival without transferring animals to new plates, probably compromising the NAC efficacy after several days. Nevertheless, by supplementing plates with 5 mM NAC (after Oh et al., Clinics, 2015 and De Magalhaes Filho et al., Nature Comm, 2018), we noticed a small but significant increase in cold survival, at least up to 9 days, which is shown in Fig. 7c.

13. Fig. 6C: The drug exposure effect on cells that were not cold-shocked but grown at 37C should be quantified here, too. It needs to be excluded that the drugs promote neurite length increase in general. The same issue applies to Fig. S6C.

This was done and we did not observe drug-induced neurite increase at 37°C (now Fig. 8c and Supplementary Fig. 6c).

14. Figure 7B and C: The effect of these drugs on cells not exposed to cold-shock should also be shown.

This was done and we did not observe any obvious effect of drugs at 37°C (now Fig. 9b and c).

15. Figure 7: Add measurements of ROS or Fe²⁺ levels in cells which overexpress Fth1, to show that Fth1 really affects ROS and Fe²⁺.

This is now shown in Fig. 9e and f. As expected, the overexpression of FTH1 decreased both the levels of Fe(II) and ROS.

Minor comments:

1. Page 8, line 2: Suddenly FTN-1 is mentioned, but it has not been introduced before. This should be corrected in the text.

We now deleted this sentence.

2. Fig. S1C: This panel should be moved to main Figure 1, near panel 1D.

According to the recommendation of another reviewer, we have rearranged a number of panels and this experiment is shown now, together with the other cold survival experiments in Fig. 2a.

3. In Fig. 4A, red colors are too similar. Please change.

This has been changed (now shown in Fig. 5d).

4. Fig S3: Please provide the number of genes in each cluster.

The comment refers to the current Fig. 4a. The numbers are now provided in the legend (76 genes/ Cluster1, 74/ cluster2, and 23/ cluster3).

Reviewer 2:

Overall this is a straightforward paper with well-designed and executed experiments showing interesting results with potentially important biomedical implications. Major concerns center on conceptual novelty of key findings and several issues that need to be addressed before being suitable for publication.

*1, Roles of *ftn-1* in iron-mediated ROS detoxification, stress resistance and longevity and its regulation by the DAF-16 pathway have been previously reported by several publications (e.g. PMID: 22445852; PMID: 22396654). Cold stress is also widely known to cause cell damage by ROS. In addition, BAM15 and other antioxidants against iron-mediated ROS have also been previously shown to increase cold survival of mammalian neurons (PMID: 29576452). It is questionable how much novelty it is conceptually or mechanistically to establish a link between ferritin to resistance of cold stress.*

We agree that some observations that we made on cold resistance were previously reported in other physiological contexts. The above-mentioned and additional papers are discussed and cited in the revised version. Admittedly, we could have done a better job describing previous studies connecting cold to ROS in mammalian cells. Those and additional papers are now cited in various parts of the manuscript: Refs. 22, 23, and 24 (Introduction); 22, 23, 51, 53, 64, and 65 (Results); and 22, 51, and 53 (Discussion). We believe that our findings strengthen and extend those earlier studies and, importantly, demonstrate their importance in intact animals.

In addition to linking and extending previous observations, we believe we made a number of original observations, which are now supported by additional data. First, we show a role for both DAF-16 and PQM-1 in promoting cold survival via the upregulation of *ftn-1*. Second, we present evidence, supported by new experiments, for the function of FTN-1 in iron detoxification, rather than storage. Third, extending our studies to neurons, we suggest a mitochondrial origin of Fe(II) (iron was not examined by Ou et al.) and demonstrate that ferritin, similar to drugs, targets the Fe(II)-ROS axis, which may have therapeutic implication. We have extensively rewrote the manuscript and, hopefully, explain now better both the background and novelty of our study.

*2, Cold stress especially under severe cold conditions like 2-4 C used in the paper is known to arrest most *C. elegans* behaviors including locomotion, pumping and feeding, causing essentially a starvation like response. This is also consistent with the known role of DAF-16 in mediating starvation response and activating *ftn-1*. It would be important to examine if the regulatory axis the author proposed mediates a specific response to cold or a general response to starvation.*

Indeed, starvation is accompanied by DAF-16 mediated upregulation of *ftn-1*. To test whether the regulatory axis that we report is specific to cold, or reflects a starvation response in cold-treated animals, we followed the expression of fluorescently tagged DAF-16 and PQM-1. DAF-16 was reported to translocate into the nucleus during short, but not long-term starvation (Henderson and Johnson, *Curr Biol*, 2001 and Weinkove et al., *BMC Biol*, 2006). As expected, we observed the nuclear DAF-16 at day 1 of starvation, but not at day 3 (Fig. 1 for Rev. 2). This is in contrast to the perduring nuclear DAF-16 in the cold. Also, while PQM-1 is also expressed and enriched in the nuclei in the cold, we saw no expression of PQM-1 during starvation, neither at day 1 nor day 3 of starvation (Fig. 1 for Rev. 2). Thus, the regulatory axis that we describe appears to be distinct from the one upregulating *ftn-1* during starvation.

3, ftn-1 has been previously reported to be regulated by several other transcription factors including HIF-1 and ELT-2 (PMID: 18024960; PMID: 22194696), in addition to DAF-16. That literature should have been at least cited and discussed. Is the phenotype described in this study dependent on HIF-1 and ELT-2?

As requested, we included now a discussion of the regulation of *ftn-1* by other TFs, like ELT-2 and HIF-1, and cite appropriate papers. It is entirely possible that these and/or additional transcription factors play a role in the regulation of *ftn-1* in the cold. However, we would like to argue that a detailed description of that regulation goes beyond the scope of this study.

4, As this study focuses on ftn-1 regulation and roles in cold resistance, it would be more appropriate to expand analysis of ftn-1 including key questions of importance on how DAF-16/PQM-1/ftn-1 might be regulated by cold, specificity of regulation in stress types, and functional consequences on different tissues especially cold stressed neurons in C. elegans. Figure 1 and 2 on less important ETS-4 and DAF-16 should be moved to supplementary data.

We agree that the cold-sensing aspect by the DAF-16/PQM-1/*ftn-1* is very interesting. In the past, we tested the potential involvement of cold-sensing TRP channel, reported to function at 15°C, but found that, at least at 4°C, the *trp-2* mutants survived cold like wt. We also tested a number of mutants with impaired sensory neurons, but they also survived cold like wt. The nature of cold sensing and potential tissue-specificity are both interesting questions. However, I hope the Reviewer will agree that they go beyond the scope of the current manuscript. As for the figure organization, we re-organized them according to comments from all reviewers and to accommodate additional data.

5, The authors conclude that FTN-1 promotes cold survival via its ferroxidase activity. Did the author test the assumption that enzymatic site mutations affect only ferroxidase activity rather than gene expression, protein abundance, protein-protein interaction or cellular localizations?

We cannot exclude all possible scenarios, for example that FeOx-dead FTN-1 fails to localize to sub-cellular compartments or engage in putative protein-protein interactions. However, as requested also by another reviewer, we performed additional experiments to characterize the effects of FeOx

mutations. We produced recombinant FTN-1 and -2, either wt or carrying the FeOx mutations (the same mutations that we previously introduced in the endogenous *ftn-1* gene). We noticed that both wt and mutated proteins were produced at similar levels and were able to form high molecular weight complexes, possibly corresponding to ferritin cages (Supplementary Fig. 4d). We followed previously published procedures to examine functions of these proteins and observed that, as predicted, FTN-1 (and -2) have FeOx activity that is abolished in the mutant proteins (Fig. 6d). Similar to previous studies, we observed that the FeOx-dead proteins bound iron (Fig. 6f), although with reduced kinetics (Fig. 6e). Finally, since we were unable to use antibodies or create functional tagged *ftn-1*, we compared the levels of wt and FeOX-dead FTN-1 (untagged), in *ets-4* animals at day 3 in the cold, by semi-quantitative mass-spectrometry, and found no obvious differences in their levels (Supplementary Fig. 4e). Thus, while the mutations abolish the FeOx activity of FTN-1 (and -2), they appear to do so without obvious distortions to the proteins.

Reviewer 3:

*Overall, the findings in this manuscript are interesting and novel in that they show a role for both DAF-16 and PQM-1 in promoting cold survival via upregulation of *ftn-1* as well as a novel function for FTN-1 in iron detoxification rather than iron sequestration. The extension of the studies to murine neurons show an important function for ferritin in cold survival in murine neurons, which may have therapeutic implications.*

Major comments

*1. For readability and clarity, some figures and sections should be reorganized. Several sections describe supplemental data that are essential for understanding the experiments, and should be included in the main body of the manuscript. The manuscript is challenging to read because the reader has to go back and forth between supplemental and main figures. Also, the manuscript requires considerable editing. In addition, some sections could be reorganized. For example, the first section of the manuscript "Inhibition of ETS-4 improves *C. elegans* survival in the cold" contains Fig. 1D-F, which show survival curves with insulin signaling mutants; however, insulin signaling is discussed in the second section "The enhanced cold survival requires both DAF-16 and PQM-1". A suggestion for reorganization of these sections/figures is described below.*

The first section containing Figs. 1A-C could also contain Figs. S1A-B. Together, these figures nicely describe cold survival data.

The second section containing Figs. S1C, S1D and Figs. 1D-F could be a new figure (Fig. 2). The data in these figures show that cold survival requires insulin signaling. New Figure 2 would include Fig. 2A (S1C), 2B (S1D), 2C(1D), 2D(1E), 2E(1F).

We are grateful for these constructive suggestions. We re-organized the figures accordingly. Fig. 1 contains now data on *ets-4* as negative regulator of cold survival, while Fig. 2 presents the data on insulin signaling and demonstrates the requirement for *daf-16* and *pqm-1*.

2. In the section "Identification of a PQM-1 and DAF-16 coregulated gene promoting cold survival", the heat map/cluster analysis and ChIP ENCODE data in Fig. S3A should be in the main body of the paper. It provides a clear visual presentation of cluster analysis, and PQM-1 and DAF-16 binding. The DAF-1 and PQM-1 like motifs in Fig. S3B could be included in Fig. 3 or in the text. The Table showing enriched motifs could remain in the supplement.

Following Reviewer's suggestion, the heat map/cluster analysis is now part of a new main figure (Fig. 4a). Also, the DAF-16 and PQM-1 like motifs are mentioned now in the text, while the table remains in the supplement (Supplementary Fig. 3a).

3. Include raw RNA-seq datasets as well as a table in the supplement showing mRNAs altered in the different strains.

The raw RNAseq dataset was submitted to GEO as mentioned in the original manuscript. The information and link to the GEO are now provided in the Data availability statement. There is also a table including the RNA counts for all genes in all the samples. Additionally, we included now Supplementary Table 4 listing changing genes in different strains.

4. What are the molecular weights of species in Peak 1 (HMW) and Peak 2 (ferritin) in Fig. S4C-D? Is it known whether FTN-1 or FTN-2 form homopolymers or heteropolymers? Also, include a statement in the legend that homogenates for these experiments are from cold-treated animals.

The comment refers to current Figs. 6a and b. In order to evaluate the size of ferritin in worm extract, we performed SEC-ICP-MS on Agilent BioSEC5 column, using the 24-mer complex of horse spleen

ferritin as size standard (theoretical molecular weight of ~440 kDa). We found that horse ferritin and Peak 2 of worm ferritin had very similar retention time (Fig. 1 for Rev 3), suggesting that also worm ferritin likely forms a 24-mer complex

Fig. 1 for Rev. 3. Native soluble iron-binding species from wild-type worm extract and horse spleen ferritin standards (100 mg/L and 500 mg/L iron) separated and detected by SEC-ICP-MS. Iron is mostly associated with high molecular weight complexes (Peak 1) and ~440 kDa ferritin (Peak 2).

(theoretical molecular weight of ~468 kDa). Loss of FTN-2 essentially abolishes Peak 2 (Fig. 6b), indicating that FTN-2 is its main component. This is in agreement with protein-bound iron analysis from *C. elegans* homogenates, performed by SEC-ICP-MS in James et al. (Chem. Sci., 2015), where Peak 2 was identified to contain FTN-2 by mass spectrometry and to be abolished in *ftn-2* mutant. Also, Hare et al. (RSC Adv., 2016) performed a molecular weight calibration of the Agilent BioSEC5 column, and found that the retention time of ferritin standard overlapped with Peak 2), similar to our results. On the other hand, our Native-PAGE experiments (present Supplementary Fig. 4d) showed that recombinant FTN-1

and -2 form large complexes of an apparent size in the range between 480-720 kDa, which exceeds the theoretically predicted molecular weight of the 24-mer (~468 kDa). This suggests that the apparent size (radius of gyration) of ferritin complex is larger than expected based on its molecular mass, which is perhaps not surprising, since ferritin cage is a hollow sphere, whose dimensions are larger than a corresponding filled-in ball of the same mass. We did not use other molecular weight standards to determine the molecular weights of Peak1 species, but they must be bigger than 468 kDa, since the retention time is shorter than ferritin.

To our knowledge, it is not known whether FTN-1 or FTN-2 form homopolymers or heteropolymers. However, human ferritins can exist both as homopolymers (homo- FTH1 or FTL) and as heteropolymers of variable FTH1:FTL ratios. Thus, it is likely that also worm FTN-1 and FTN-2 can form homo- and heteropolymers, whose exact composition depends on the relative availability of individual monomers.

As requested, we now included a statement in the legend that these experiments were performed on cold-treated animals.

5. As shown in Fig. S4B, *ftn-1* mutants show sensitivity to iron and cold. Authors should assess *ftn-2* mutant viability under cold and iron stress similar to *ftn-1* experiments.

This comment concerns the FAC experiments, where animals were challenged with extraneous iron. These experiments were performed at an early stage, when the goal was to test whether iron may play a role in cold survival. Subsequent experiments showed that to be the case but also demonstrated that FTN-1 promotes cold survival without increasing the levels of total body iron, which is presumably what happens upon FAC supplementation. Thus, after careful consideration, we decided to remove these experiments from the revised manuscript. We hope the Reviewer will agree.

6. Elaborate on the model for FTN-1 “iron detoxification” vs sequestration. What is the fate of the detoxified iron - is it sequestered in FTN-2? SEC-ICP-MS plots in Fig. S4C-D could be shown in Fig.4.

We and others (cited in the manuscript) showed that FTN-2 is responsible for the bulk of iron stored in worms. Since neither the loss nor overexpression of FTN-1 appears to impact the total levels of stored iron (also in the cold), we suggest that, rather than simply sequestering iron away, FTN-1 is playing a role in iron detoxification, i.e., the conversion of ferrous iron into its ferric form. This does

not mean that FTN-1 does not bind iron. Indeed, in agreement with previous studies on human ferritins (Levi et al., *Biochemistry*, 1989 and Lawson et al., *FEBS Lett.*, 1989), our new experiments suggest that both wt and FeOx-dead FTN-1 can bind and store ferric iron. However, when ferrous ions are the source of iron, the FeOX activity speeds up iron binding and storage. We speculate that FTN-1, through its FeOx activity, facilitates conversion of ferrous into ferric iron, which then remains, at least temporarily, stored within FTN-1, thus efficiently removing the ‘toxic’ ferrous iron from the surrounding. The model does not explain why the inducible FTN-1 is used in addition to the constitutively expressed FTN-2, which is also ferroxidase active. Among possible scenarios, the de-novo produced ferritin can be more effective than the pre-existing, iron-charged ferritin, in detoxifying iron, or FTN-1 may play a role in specific cells or subcellular locations, etc.

As for testing the fate of detoxified iron, we do not see a way of tracing of iron species in this model.

As suggested, we moved now the SEC-ICP-MS plots from supplemental to main figure (currently in Fig. 6a and b).

7. Include a model for the insulin-IGF-1 signaling pathway-mediated regulation of ferritin during cold stress. How do PQM-1 and DAF-16 coregulate ftn-1?

As requested also by another reviewer, we included now a model presented in Fig. 10.

8. Cite reference related to ferritin protection from cold injury in endothelial cells. M. A. J. Zieger and Mahes P. Gupta, Hypothermic preconditioning of endothelial cells attenuates cold-induced injury by a ferritin-dependent process. Free Radical Biology & Medicine 46 (2009), 680-691.

We thank the Reviewer for pointing this out. This paper is now cited (ref. 64).

Other comments

1. Fig. 1D: for consistency label y-axis “Viability”

We re-labelled it as suggested, now in Fig. 2c.

2. For survival curves, explain “n”. Are they independent biological replicates? If so, explain biological replicates.

An individual biological replicate of survival curve has been now defined in Methods: ‘All experiments were performed in 3-4 independent biological replicates, defined as experiments performed on different days, using separate batches of nematodes. Several hundred animals (200-800) were used for scoring viability at each time point of an individual survival curve.’ It is also described in Supplementary Table 2.

*3. Lines 173-175: add (Fig. 2A, C) at the end of the sentence. From statistics in Fig. 2C, these data show significance (**p), although it is stated, “possibly with minimal increase”.*

We now refer in the text to data in our current Figs. 3a and c. We deleted the word ‘possibly’.

4. Lines 169-171: add references

Appropriate references are now included.

5. Line 195: sentence is out of context (Fig. 3C)

This sentence is now removed.

6. Lines 180-182: add (Fig. S1C)

We inserted reference to the data in current Fig. 2a in the text.

7. Line 234: add Tepper et al. reference number

This reference is now inserted.

8. Lines 234-236: List the seven genes in the text in parentheses

All seven genes are now listed in the Results.

9. Lines 279-281: Change Figs. 4B, S4C, E to (Figs. 4B, S4C, S4E).

This has been corrected. Due to rearrangement of panels and figures, the relevant data are now shown as Figs. 6a (former 4b and S4c, joint together) and Supplementary Fig. 4a (former S4e).

10. Fig. S3B: line 67-68: clarify “start -1500 end 1500-p6”

The following phrase: ‘-start -1500, -end 1500, -p 6’ that appears in the legend to current Supplementary Fig. 3a, is referring to input parameters and their actual values used in the HOMER program.

11. Line 508: “reproductively” should be “reproducibly” in Fig. 3 legend

This has been corrected.

12. Line 540: delete “Comparing”

The legends has been modified extensively for this and other figures, and the word “Comparing’ has been deleted.

13. Line 291: add the amino acid changes for ferroxidase point mutations at end of sentence.

The amino acid changes has now been written explicitly in the text, and additionally indicated graphically in Fig. 6c.

14. Line 310-311: insert Fig. 5C

Reference to data in current Fig. 7d has been inserted in the text.

15. Include SOD-5::GFP images (Fig. S5) in Fig. 5.

We have now included former Fig. S5d as current Fig. 7e.

16. Fig. S5A: modify y-axis to visualize wt levels.

The Y-axis in current Supplementary Fig. 5a has been modified to visualize wt levels.

17. Line 324: correct spelling to applicable

This has been corrected.

18. Lines 344-345: reference is needed “Remarkably ... lower cold resistance”

The reference has been added.

19. Line 348: define BAM and PI; add Ou reference

BAM15 and PI are now defined in the text, and the reference is inserted.

20. In Fig. 6A, add the concentration of DFO, BAM15 and PI and incubation time in figure legend

The required information has been added to the current Fig. 8a.

21. Fig. 6C: curves need better definition

We suspect the Reviewer refers to the description of experimental conditions. These are now changed to better describe the experimental setup.

22. Lines 418-419: Include references for antioxidant therapy in humans

The appropriate references have now been added.

REVIEWER COMMENTS

Reviewer #1 (Remarks to the Author):

I am satisfied with how the authors addressed most of my comments. The only exceptions are the following two points:

2) Thanks for the encouraging result. Please include Fig 1 for Rev 1 as a supplement figure.

6.2) Maybe my instructions were not clear enough, but what is needed here in addition to the new data provided is some proper functionality testing of the new GFP fusions. For example, does having the new DAF-16::GFP instead of wild-type DAF-16 still allow for dauer formation or longevity in *daf-2* mutants? And does having the new PQM-1::GFP instead of wild-type PQM-1 still allow for timely dauer exit or full longevity in *daf-2* mutants?

Reviewer #2 (Remarks to the Author):

The authors have addressed most concerns from this reviewer except the point regarding the role of starvation in regulating DAF-16 and *ftn-1* during cold treatment. The low quality data (Fig. 1 for Rev. 2: only single image was shown with signals not as clear as in Fig. 3a, and no statistics) on DAF-16::GFP upon starvation are not convincing. Perduring nuclear DAF-16 during cold does not exclude a contribution from starvation. In fact, Fig. 1 for Rev. 2 supports that starvation might contribute to the early phase of cold induced DAF-16 activation especially if acute cold treatment does not activate DAF-16.

Reviewer #3 (Remarks to the Author):

The authors have addressed most of my concerns.

Although ferritin iron sequestration and promotion of cold survival has been reported in mammalian cells, a novel finding in the manuscript is that *ftn-1* cold induction depends on both PQM-1 and DAF-16 localization to the same region of the *ftn-1* promoter that contains a DAE, a known PQM-1 binding site. Previous studies show that PQM-1 and DAF-16 nuclear localization is mutually exclusive.

One concern regards the model figure. The model shown does not convey the interesting results of the project. A well-designed model figure is useful for readers to quickly grasp the purpose and results of the research. The authors describe several stories in their manuscript including genetic studies examining the insulin signaling components, *pqm-1* and *daf-16*, in cold survival of *ets-4* mutants; functional genomics that identify *ftn-1* as candidate promoting cold survival; dependence of *ftn-1* cold induction both PQM-1 and DAF-16; and studies showing that FTN-1 promotes cold survival via its

ferroxidase activity. These results should be better integrated into their model. The model suggests that PQM-1 and DAF-16 bind DNA at different regions, which is misleading. As the authors suggest, DAF-16 may be recruited to DNA via PQM-1 (e.g. indirect vs direct mechanisms). This idea could be incorporated into their model. The images of faces on the alive and dead cells should be removed.

We are happy that the revision satisfied the Reviewers overall and would like to thank them for their efforts. Below is our point-by-point response to the remaining comments.

Reviewer 1:

I am satisfied with how the authors addressed most of my comments. The only exceptions are the following two points:

2) Thanks for the encouraging result. Please include Fig 1 for Rev 1 as a supplement figure. The figure is now included as supplementary Fig. S5e.

6.2) Maybe my instructions were not clear enough, but what is needed here in addition to the new data provided is some proper functionality testing of the new GFP fusions. For example, does having the new DAF-16::GFP instead of wild-type DAF-16 still allow for dauer formation or longevity in *daf-2* mutants? And does having the new PQM-1::GFP instead of wild-type PQM-1 still allow for timely dauer exit or full longevity in *daf-2* mutants?

We indeed understood the instructions differently. We used these strains exclusively to examine the localization patterns of both proteins in the cold. As shown in the previous response letter, the localization patterns observed with our and published strains were similar. Therefore, we believe that the existing data supports the nuclear localization of both proteins in the cold. Nonetheless, we performed now some functional characterization. Since the Reviewer asked about dauers or longevity, we chose the former as less time-demanding. As requested, we tested dauer entry/exit of both strains, +/- *daf-2* (Table 1 for Reviewer 1). The tagging of *daf-16* or *pqm-1* had no obvious effect on the dauer entry of the temperature-sensitive *daf-2(e1370)* mutant, which is used in such studies. As for the dauer exit, we observed a slightly better recovery of the tagged *daf-16* animals. The recovery of tagged *pqm-1* animals and non-tagged animals appeared similar. Summarizing, while we cannot exclude that the tagging may have some impact on the functionality of DAF-16 and/or PQM-1, we have no evidence of major functional impairments.

C. elegans strain	% dauer formation	% dauer exit
daf-2(e1370)	100 % (187/187)	79 % (147/187)
daf-2(e1370); daf-16::gfp-flag	100 % (360/360)	91% (327/360)
daf-2(e1370); pqm-1::mcherry-myc	100 % (208/208)	79 % (164/208)

Table 1 for Reviewer 1. To test dauer formation, about 50-100 eggs from indicated *C. elegans* strains were plated on separate *E.coli* OP50-containing NG 2% agar plates and incubated at 25°C (restrictive temperature for the *e1370* allele of *daf-2*) for 48 hours. The dauers formed on each plate were counted and are expressed as the percentage of all animals present on the plate. To test the exit from dauers, the plates containing dauers were shifted to 15°C and incubated for 48 hours. The animals exiting the dauer stage were counted and are expressed as the percentage of all animals present on the plate. Dauer formation and exit was performed in 3 biological replicates for each strain and the final results are reported as the sum of individual experiments.

Reviewer 2:

The authors have addressed most concerns from this reviewer except the point regarding the role of starvation in regulating DAF-16 and *ftn-1* during cold treatment. The low quality data (Fig. 1 for Rev. 2: only single image was shown with signals not as clear as in Fig. 3a, and no statistics) on DAF-16::GFP upon starvation are not convincing. Perduring nuclear DAF-16 during cold does not exclude a contribution from starvation. In fact, Fig. 1 for Rev. 2 supports that starvation might contribute to the early phase of cold induced DAF-16 activation especially if acute cold treatment does not activate DAF-16.

The figure previously submitted for Reviewer 2 was meant as a simple illustration that DAF-16 and PQM-1 behave differently in the cold than upon starvation. For DAF-16, the difference is in the length of its nuclear expression. Consistent with the published data (Henderson and Johnson, *Curr Biol*, 2001; Weinkove et al., *BMC Biol*, 2006), we observed the nuclear DAF-16 upon short but not long-term starvation. This contrasts with the perduring nuclear expression of DAF-16 in the cold. However, the main

difference was in the expression of PQM-1. PQM-1 is nuclear in the cold (shown in our study) but we observed no expression of PQM-1 in starved animals. To illustrate this better, please see the new figure below of a better quality (Fig. 1 for Rev. 2). Since PQM-1 is not detectable in starved animals, there was no signal to quantify.

Because DAF-16 is nuclear in both starved (transiently) and cold-treated (continuously) animals, the Reviewer suggested that "starvation might contribute to the early phase of cold induced DAF-16 activation". Please note that, even if the upregulation of DAF-16 in the initial phase of cold treatment is related to starvation, it is insufficient to explain the expression of *ftn-1*. This is because the *ftn-1* expression in cold-treated animals also requires PQM-1, which in starved animals is not expressed (see the figure). Thus, our results do suggest that the regulatory axis upregulating *ftn-1* in the cold is distinct from the one operating during starvation.

Figure 1 for Reviewer 2. Shown are representative fluorescence (mCherry) and light (DIC) images. Animals were grown at 20°C on *E. coli* OP50-containing NG 2% agar plates for at least 2 generations before starvation. Late L3/early L4 larvae were picked to NG 2% plates without bacteria and kept at 20°C for 24 hours. Arrowheads point to nuclei. Note that neither the fed nor starved animals expressed PQM-1::mCherry in the nucleus. Background signal comes from auto-fluorescent gut granules. The images were taken with Zeiss Imager Z1 fluorescence microscope. Scale bar: 10 µm.

Reviewer 3:

The authors have addressed most of my concerns.

*Although ferritin iron sequestration and promotion of cold survival has been reported in mammalian cells, a novel finding in the manuscript is that *ftn-1* cold induction depends on both PQM-1 and DAF-16 localization to the same region of the *ftn-1* promoter that contains a DAE, a known PQM-1 binding site. Previous studies show that PQM-1 and DAF-16 nuclear localization is mutually exclusive.*

*One concern regards the model figure. The model shown does not convey the interesting results of the project. A well-designed model figure is useful for readers to quickly grasp the purpose and results of the research. The authors describe several stories in their manuscript including genetic studies examining the insulin signaling components, *pqm-1* and *daf-16*, in cold survival of *ets-4* mutants; functional genomics that identify *ftn-1* as candidate promoting cold survival; dependence of *ftn-1* cold induction both PQM-1 and DAF-16; and studies showing that FTN-1 promotes cold survival via its ferroxidase activity. These results should be better into integrated into their model. The model suggests that PQM-1 and DAF-16 bind DNA at different regions, which is misleading. As the authors suggest, DAF-16 may be recruited to DNA via PQM-1 (e.g. indirect vs direct mechanisms). This idea could be incorporated into their model. The images of faces on the alive and dead cells should be removed.*

As suggested, we now removed the "faces" from the figure and modified the drawing, to allow for an indirect binding of DAF-16 to the *ftn-1* promoter. The players mentioned by the Reviewer are included in the model. As indicated by the Reviewer, the manuscript describes several stories and it is difficult to incorporate all results in one model. Nonetheless, we hope the Reviewer will agree that the corrected model sufficiently summarizes the key findings.

REVIEWERS' COMMENTS

Reviewer #1 (Remarks to the Author):

I am now fully satisfied with the revisions.

Reviewer #2 (Remarks to the Author):

The revised paper has been improved and addressed previous concerns.

REVIEWERS' COMMENTS

Reviewer #1 (Remarks to the Author):

I am now fully satisfied with the revisions.

Reviewer #2 (Remarks to the Author):

The revised paper has been improved and addressed previous concerns.

Thank you!